# Efficient Data Selection at Scale
# via Influence Distillation

**Mahdi Nikdan**[*][†]
ISTA & Google Research

**Vincent Cohen-Addad**[†]
Google Research

**Dan Alistarh**
ISTA & Red Hat AI

**Vahab Mirrokni**
Google Research

## Abstract

Effective data selection is critical for efficient training of modern Large Language Models (LLMs). This paper introduces Influence Distillation, a novel, mathematically-justified framework for data selection that employs second-order information to optimally weight training samples. By distilling each sample's influence on a target distribution, our method assigns model-specific weights that are used to select training data for LLM fine-tuning, guiding it toward strong performance on the target domain. We derive these optimal weights for both Gradient Descent and Adam optimizers. To ensure scalability and reduce computational cost, we propose a *landmark-based approximation*: influence is precisely computed for a small subset of "landmark" samples and then efficiently propagated to all other samples to determine their weights. We validate Influence Distillation by applying it to instruction tuning on the Tulu V2 dataset, targeting a range of tasks including GSM8k, SQuAD, and MMLU, across several models from the Llama and Qwen families. Experiments show that Influence Distillation matches or outperforms state-of-the-art performance while achieving up to $3.5\times$ faster selection.

## 1 Introduction

The rise of Large Language Models (LLMs) has driven significant advances in natural language processing; yet, training and fine-tuning these models requires massive computational resources and carefully-curated datasets. One key direction towards improved training efficiency has been via *data selection and data weighting methods* [Xia et al., 2024, Yin and Rush, 2024, Antonello et al., 2020, Marion et al., 2023, Ankner et al., 2024, Li et al., 2023a, Ivison et al., 2025, Axiotis et al., 2024, Xie et al., 2023a, Engstrom et al., 2024a, Huang et al., 2024, Feng et al., 2024], which aim to curate training subsets that maximize a model's effectiveness, often with respect to a particular *target data distribution* or downstream task. However, existing approaches typically rely on heuristics—such as perplexity-based filtering—or require expensive proxy model training or expensive embedding functions to generate data representations.

More precisely, existing methods face several limitations. First, many existing methods utilize fixed, model-agnostic features or representations (e.g., static embeddings) that may not capture the full relationship between training samples and the target distribution [Yin and Rush, 2024, Antonello et al., 2020, Marion et al., 2023, Ankner et al., 2024]. Second, methods that update weights during training lack theoretical justification and can be unstable [Xie et al., 2023a, Huang et al., 2024]. Lastly, approaches that rely on reference model training or costly embeddings are computationally

---

[*]Work done while an intern at Google Research.

[†]Correspondence to mnikdan@ista.ac.at and cohenaddad@google.com.

intensive and often challenging to scale [Li et al., 2023a, Xia et al., 2024, Ivison et al., 2025, Yu et al., 2024, Feng et al., 2024]. Thus, there remains a clear need for a mathematically-grounded, efficient, and scalable framework for data selection that directly optimizes for performance on a specific target distribution.

**Contribution.** We introduce *Influence Distillation*, a novel framework for data selection that addresses these challenges. Given a pre-trained model and a target task (represented by a small target dataset), Influence Distillation formulates the influence of training samples on the target distribution's loss via a second-order approximation. Influence Distillation directly optimizes sample weights by analyzing how each training sample, if included in a gradient step, is expected to affect model performance on the target data. This formulation leads to a quadratic optimization objective for the sample weights, which we demonstrate can be solved efficiently. We provide derivations for these optimal weights under both standard Gradient Descent and the adaptive Adam optimizer, backed by theoretical justifications.

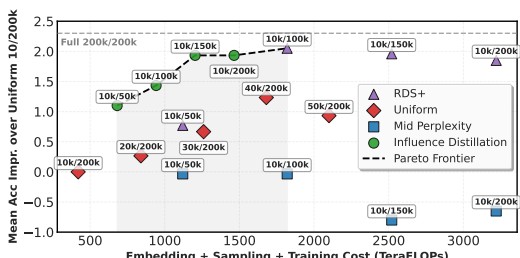

Figure 1: Average improvement over uniform sampling across six tasks vs. runtime. The model used is Llama2-7B [Touvron et al., 2023], and the training dataset is Tulu V2 [Ivison et al., 2023]. The annotation "M/N" indicates that the method selected M samples from a pool of size N. Further details are provided in Section 5.

To ensure scalability to large datasets, we further introduce an efficient *landmark-based* approximation. This approach first involves selecting a small subset of "landmark" samples and precisely computing their influence. The influence for all other samples is then efficiently approximated by transferring the computed influence from these landmarks. This transfer mechanism is guided by a novel and computationally inexpensive embedding space derived from Jacobian-vector Products. This significantly reduces the computational overhead of gradient computations for the entire dataset.

We validate Influence Distillation via comprehensive instruction tuning experiments using standard open LLMs (from the Llama [Touvron et al., 2023, Grattafiori et al., 2024] and Qwen [Team, 2024] families) on the Tulu V2 [Ivison et al., 2023] training dataset, while targeting advanced downstream tasks like MMLU [Hendrycks et al., 2021a,b], mathematics and code. Our results demonstrate that Influence Distillation not only substantially outperforms uniform random selection but also in most cases matches or exceeds the performance of state-of-the-art data selection methods, while offering significant computational speedups on the same selection problem—up to $3.5\times$ in embedding + selection runtime. This positions Influence Distillation as a strong method on the Pareto frontier of overall embedding, selection and training cost versus downstream task accuracy (see Figure 1).

## 2 Related Work

Data selection ('pruning') and weighting methods have become increasingly important in the context of efficient LLM training. In a celebrated paper, Sorscher et al. [2022] et al. showed that (model-agnostic) data pruning, and in particular deduplication, helps go beyond scaling laws for LLMs. This was later further improved by Abbas et al. [2023].

Early work on model-dependent data pruning focused on heuristics like perplexity-based filtering and confidence-based selection: Marion et al. [2023] found that selecting examples with moderate perplexity scores often outperforms training on the full dataset or examples selected by other metrics. Do and Gaspers [2019] introduced DSIR, which uses importance resampling based on n-gram features to select relevant training examples, with promising results on mathematical reasoning and clinical text summarization. Similarly, Xie et al. [2023b] proposed clustering loss trajectories to identify representative training examples, though their approach focused more on general domain adaptation rather than specific target distributions. Another approach, so-called Classifier, was introduced by Brown et al. [2020] and has been employed in subsequent work (Gao et al. [2020], Chowdhery et al. [2023], Du et al. [2022]. Other strategies include selecting examples that maximize the loss difference between LMs trained on candidate and reference datasets (Moore and Lewis

[2010], Axelrod [2017], Feng et al. [2022]). Another direction is employing large, well-trained language models to select informative samples [Chen et al., 2023a, Pang et al., 2024]. Simpler, yet common, techniques involve filtering documents based on length or the presence of excessive special characters (Raffel et al. [2020], Xie et al. [2023b]). A related, though distinct, task in the LM domain is optimizing the weights for sampling from mixed data sources (Chen et al. [2024], Albalak et al. [2023]). Recently, Ivison et al. [2025] proposed RDS+, which uses similarity between model-dependent embeddings computed by a position-weighted mean pool of the last hidden layer states.

Recent work has also highlighted the importance of considering the training dynamics when selecting data. Zhou et al. [2023a] proposed measuring "learnability" based on loss changes during training, while Swayamdipta et al. [2020] introduced "dataset cartography" to analyze training dynamics across examples. These methods provide useful signals about which examples are most valuable for training; at the same time, they require training reference models which can be computationally expensive. For large-scale applications, Bhatt et al. [2024] evaluated various data selection approaches for LLM fine-tuning, and found that facility-location selection based on hidden representations was particularly effective. However, Tirumala et al. [2023] observed that generating these representations for large datasets remains computationally challenging. More recently, Engstrom et al. [2024b] framed the data selection problem as an optimization problem: Given the learning algorithm, find the subset of the data that maximizes the performance of the trained model. To obtain an efficient solution, they design a model that given a subset of the training data $S$ and a target example $t$, predicts the loss of the model trained on $S$ on $t$. Axiotis et al. [2024] recently use coreset-related ideas to propose a computationally efficient way of sampling an unbiased estimator of the model loss from the training data so as to train on a smaller input.

While previous methods like DSIR and facility location selection rely on fixed features or representations, our method directly optimizes sample weights based on their influence on the target distribution through a second-order approximation. Importantly, this does not require training proxy model to predict the value of the elements and is computed directly from the input, model and learning algorithm. Unlike curriculum learning or confidence-based approaches that update weights during training, we derive optimal weights analytically for both SGD and Adam optimizers. In contrast to methods that require training reference models, our landmark-based approximation allows efficient weight computation without extensive pre-training.

There is a large body of work on data selection methods for other learning tasks and mode, and it is beyond the scope of this paper to provide a detailed overview. We refer the reader to Kaushal et al. [2019], Killamsetty et al. [2021], Wei et al. [2015], Chen et al. [2023b], Cao et al. [2023], Sener and Savarese [2017] and references therein.

# 3 Method

## 3.1 Problem and Notation

Let $\boldsymbol{\theta} \in \mathbb{R}^d$ be the model parameters. For any dataset $D$ of size $n$ and any vector of sample weights $\boldsymbol{w} = [w_1, w_2, ..., w_n]^T$, denote $\mathcal{L}(\boldsymbol{\theta}; D, \boldsymbol{w}) = \frac{1}{n}\sum_{i=1}^{n} w_i \, \ell(\boldsymbol{\theta}; D_i)$ as the weighted average of the model loss $\ell$ on the samples of dataset $D$ at point $\boldsymbol{\theta}$. Additionally, define $\mathcal{M}(\boldsymbol{\theta}; D, \boldsymbol{w})$ as a training mechanism that returns the parameters after being trained on a dataset $D$ weighted by $\boldsymbol{w}$. Unless otherwise stated, we will assume $\mathcal{M}$ is simply one step of (full) gradient descent.

Let $\mathcal{S}$ and $\mathcal{T}$ represent the training (source) and downstream (target) distributions, respectively. Assume we have access to a dataset $S$ sampled from $\mathcal{S}$ and a small representative dataset $T$ from $\mathcal{T}$. Our high-level goal will be to determine sample weights $\boldsymbol{w}^*$ such that:

$$\boldsymbol{w}^* = \arg\min_{\boldsymbol{w}} \ \mathcal{L}(\mathcal{M}(\boldsymbol{\theta}; S, \boldsymbol{w}); T, \mathbb{1}) \tag{1}$$

where $\mathbb{1} \in \mathbb{R}^{|T|}$ represents the all-ones vector. In words, we wish to find sample weights $\boldsymbol{w}$ for instances within the source dataset $S$, such that training on $S$ using these weights results in minimal loss on the target dataset $T$. Notably, this notation also allows for the special case of $\mathcal{S} = \mathcal{T}$, where our method would find weights that maximize in-distribution loss improvement.

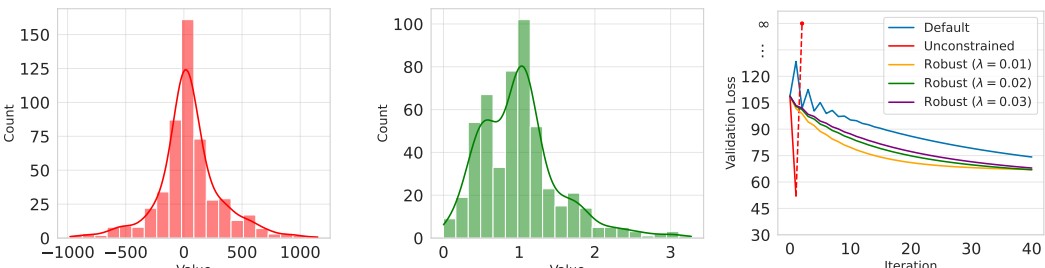

Figure 2: (Left) Distribution of unconstrained weights, (Middle) Distribution of robust weights for $\lambda = 0.02$, and (Right) validation loss during training with different variants in the running experiment setting. Robust weights are found by minimizing Objective 7 using the SLSQP algorithm [Kraft, 1988] implemented in the SciPy library [Virtanen et al., 2020].

## 3.2 A Running Example

Throughout this section, we utilize a toy training setting to illustrate variants of our method. Specifically, we consider a linear regression model parameterized by $\boldsymbol{\theta}$ with the loss function $\ell(\boldsymbol{\theta}; \mathbf{x}, y) = (\boldsymbol{\theta}^T \mathbf{x} - y)^2$ for any $\boldsymbol{\theta}, \mathbf{x} \in \mathbb{R}^d, y \in \{0, 1\}$. For the source dataset, we sample 256 random instances from the first two classes of the CIFAR-10 dataset [Krizhevsky, 2009] and combine them with 256 synthetic samples generated from a Gaussian distribution with the same mean and standard deviation as the real samples. The target dataset consists of another set of 256 samples from CIFAR-10. We use gradient descent with a learning rate of $10^{-3}$ as the optimizer. Finally, the loss values are reported on a validation dataset of size 256, also sampled from CIFAR-10.

## 3.3 Influence Distillation

**Case 1: Unconstrained Weights.** Let $\mathbf{g}_T(\boldsymbol{\theta}) = \nabla_{\boldsymbol{\theta}} \mathcal{L}(\boldsymbol{\theta}; T, \mathbb{1})$ and $\mathbf{H}_T(\boldsymbol{\theta}) = \nabla_{\boldsymbol{\theta}}^2 \mathcal{L}(\boldsymbol{\theta}; T, \mathbb{1})$ denote the gradient vector and Hessian matrix of the loss with respect to the model parameters on the target dataset. Construct $\mathbf{G}_S(\boldsymbol{\theta}) \in \mathbb{R}^{|S| \times d}$ by stacking the gradients of the loss with respect to $\boldsymbol{\theta}$ across samples of $S$. As mentioned before, assume $\mathcal{M}$ is one step of gradient descent, i.e., $\mathcal{M}(\boldsymbol{\theta}; \mathcal{D}, \boldsymbol{w}) = \boldsymbol{\theta} - \eta \nabla_{\boldsymbol{\theta}} \mathcal{L}(\boldsymbol{\theta}; D, \boldsymbol{w}) = \boldsymbol{\theta} - \frac{\eta}{|S|} \mathbf{G}_S^T(\boldsymbol{\theta}) \boldsymbol{w}$, where $\eta$ denotes the learning rate. We estimate Objective 1 by:

$$\boldsymbol{w}^* = \arg\min_{\boldsymbol{w}} \ \mathcal{L}(\mathcal{M}(\boldsymbol{\theta}; S, \boldsymbol{w}); T, \mathbb{1})$$

$$= \arg\min_{\boldsymbol{w}} \ \mathcal{L}(\boldsymbol{\theta} - \frac{\eta}{|S|} \mathbf{G}_S^T(\boldsymbol{\theta}) \boldsymbol{w}; T, \mathbb{1})$$

$$\approx \arg\min_{\boldsymbol{w}} \ [\mathcal{L}(\boldsymbol{\theta}; T, \mathbb{1}) - \frac{\eta}{|S|} \mathbf{g}_T^T(\boldsymbol{\theta}) \mathbf{G}_S^T(\boldsymbol{\theta}) \boldsymbol{w} + \frac{\eta^2}{2 |S|^2} \boldsymbol{w}^T \mathbf{G}_S(\boldsymbol{\theta}) \mathbf{H}_T(\boldsymbol{\theta}) \mathbf{G}_S^T(\boldsymbol{\theta}) \boldsymbol{w}]$$

$$= \arg\min_{\boldsymbol{w}} \ [-\mathbf{g}_T^T(\boldsymbol{\theta}) \mathbf{G}_S^T(\boldsymbol{\theta}) \boldsymbol{w} + \frac{\eta}{2 |S|} \boldsymbol{w}^T \mathbf{G}_S(\boldsymbol{\theta}) \mathbf{H}_T(\boldsymbol{\theta}) \mathbf{G}_S^T(\boldsymbol{\theta}) \boldsymbol{w}] \qquad (2)$$

where the approximation comes from a second-order Taylor expansion, i.e., $\mathcal{L}(\boldsymbol{\theta} + \boldsymbol{\delta}; T, \mathbb{1}) \approx \mathcal{L}(\boldsymbol{\theta}; T, \mathbb{1}) + \mathbf{g}_T^T(\boldsymbol{\theta}) \boldsymbol{\delta} + \frac{1}{2} \boldsymbol{\delta}^T \mathbf{H}_T(\boldsymbol{\theta}) \boldsymbol{\delta}$ where $\boldsymbol{\delta}$ is replaced with $-\frac{\eta}{|S|} \mathbf{G}_S^T(\boldsymbol{\theta}) \boldsymbol{w}$.

Next, we define two key objects, $\mathbf{p} \in \mathbb{R}^{|S|}$ and $\mathbf{Q} \in \mathbb{R}^{|S| \times |S|}$, as follows:

$$\mathbf{p}(\boldsymbol{\theta}; S, T) = \mathbf{G}_S(\boldsymbol{\theta}) \mathbf{g}_T(\boldsymbol{\theta}), \qquad (3)$$

$$\mathbf{Q}(\boldsymbol{\theta}; S, T) = \frac{1}{|S|} \mathbf{G}_S(\boldsymbol{\theta}) \mathbf{H}_T(\boldsymbol{\theta}) \mathbf{G}_S^T(\boldsymbol{\theta}), \qquad (4)$$

For brevity, unless stated otherwise, we will omit $S$ and $T$ from the arguments of $\mathbf{p}$ and $\mathbf{Q}$. Let

$$f(\boldsymbol{w}; \boldsymbol{\theta}) = -\mathbf{p}(\boldsymbol{\theta})^T \boldsymbol{w} + \frac{\eta}{2} \boldsymbol{w}^T \mathbf{Q}(\boldsymbol{\theta}) \boldsymbol{w}. \qquad (5)$$

Then, the objective in Equation 2 becomes

$$\boldsymbol{w}^* = \arg\min_{\boldsymbol{w}} \ f(\boldsymbol{w}; \boldsymbol{\theta}). \qquad (6)$$

In words, $f$ represents a scaled approximation of the change in loss on $T$ when the model at point $\boldsymbol{\theta}$ is trained on $S$ with weights $\boldsymbol{w}$. It is a quadratic function in $\boldsymbol{w}$, as $\mathbf{p}$ and $\mathbf{Q}$ do not depend on $\boldsymbol{w}$. This objective can be minimized in closed form as $\boldsymbol{w}^* = \frac{1}{\eta}\mathbf{Q}(\boldsymbol{\theta})^{-1}\mathbf{p}(\boldsymbol{\theta})$.

**Discussion.** While simple, the proposed solution has several crucial limitations: (a) it may produce negative or highly irregular sample weights, such as excessively large values, which lack intuitive interpretation, (b) the weights may overfit to the current set of parameters $\boldsymbol{\theta}$, and (c) the weights may also overfit to the target dataset. The first two issues can be easily observed in our running experiment. The irregularity of the weights is illustrated in Figure 2 (left). Furthermore, Figure 2 (right) demonstrates that unconstrained weights become invalidated after just one step of training, suggesting that the weights "overfit" to the current model parameters $\boldsymbol{\theta}$. We note that, since the model in our running example is linear, the second-order approximation is exact—thus, the first update step reaches the optimum on $T$. However, this behavior does not generalize to non-linear models.

**Case 2: Robust Weights.** We modify Objective 6 to address the above limitations. First, we restrict the weights to non-negative values, i.e., $\forall\, 1 \leq i \leq |S|\colon \boldsymbol{w}_i \geq 0$. Second, we require the weights to sum to the size of the source dataset, $\boldsymbol{w}^T\mathbb{1} = |S|$. This prevents weights from becoming excessively large and ensures that rescaling the weights does not change the effective step size: using $\alpha\boldsymbol{w}$ with learning rate $\eta$ is equivalent to using $\boldsymbol{w}$ with learning rate $\alpha\eta$.

To mitigate "overfitting", a standard approach is to add a regularization term. Indeed, Appendix B derives such a term for linear models. In the general case, we employ a simple L2 regularization term.

**The Robust Influence Objective.** Hence, we define the robust Influence Distillation objective:

$$\boldsymbol{w}^* = \underset{\boldsymbol{w}}{\arg\min}\ f(\boldsymbol{w};\boldsymbol{\theta}) + \frac{\lambda}{2}\|\boldsymbol{w}\|_2^2,\ \ s.t.\ \begin{cases} \boldsymbol{w} \geq 0 \\ \boldsymbol{w}^T\mathbb{1} = |S| \end{cases} \tag{7}$$

Refer to Section 4.4 for a discussion on how we tune $\lambda$ in practice.

We compute the robust weights with $\lambda \in \{0.01, 0.02, 0.03\}$ in the context of our running example. Figure 2 (right) highlights the effectiveness of these robust weights, showing that all three configurations outperform the default weights while remaining stable throughout training. Additionally, Figure 2 (middle) depicts the distribution of weights for $\lambda = 0.02$.

**Adam Optimizer.** The Adam optimizer [Kingma, 2014] is the default choice for fine-tuning LLMs. Therefore, we tailor our method for Adam optimizers. To this end, we employ a greedy approach, where we assume the first- and second-order momentums ($\mathbf{m}$ and $\mathbf{v}$, respectively) are fixed after a warm-up. In this case, the $\mathbf{Q}^{\text{Adam}}$ and $\mathbf{p}^{\text{Adam}}$ objects are calculated as follows:

$$\mathbf{p}^{\text{Adam}}(\boldsymbol{\theta}) = \mathbf{G}_S^{\text{Adam}}(\boldsymbol{\theta})(\mathbf{g}_T(\boldsymbol{\theta}) - \frac{\eta}{|S|}\,\mathbf{H}_T(\boldsymbol{\theta})\mathbf{b}), \tag{8}$$

$$\mathbf{Q}^{\text{Adam}}(\boldsymbol{\theta}) = \frac{1}{|S|}\mathbf{G}_S^{\text{Adam}}(\boldsymbol{\theta})\mathbf{H}_T(\boldsymbol{\theta})\mathbf{G}_S^{\text{Adam}}(\boldsymbol{\theta})^T, \tag{9}$$

where $\mathbf{b} = \frac{\beta_1\mathbf{m}}{(1-\beta_1^s)(\sqrt{\frac{\mathbf{v}}{1-\beta_2^s}}+\epsilon)}$, and $\mathbf{G}_S^{\text{Adam}}(\boldsymbol{\theta})$ is constructed by element-wise multiplying every row of $\mathbf{G}_S(\boldsymbol{\theta})$ by $\mathbf{a} = \frac{1-\beta_1}{(1-\beta_1^s)(\sqrt{\frac{\mathbf{v}}{1-\beta_2^s}}+\epsilon)}$. Additionally, $s$ is the number of warm-up steps, and $(\beta_1, \beta_2, \epsilon)$ are Adam hyperparameters. See Appendix C for more details.

**Handling Variable Lebel Lengths.** A common practice in data selection is to normalize the gradients prior to measuring similarities [Xia et al., 2024]. This is motivated by the observation that the norm of a sample's gradient tends to correlate negatively with the number of label tokens, thereby biasing unnormalized gradient-based methods toward shorter samples. Normalizing the gradients mitigates this issue and shifts the similarity measure from a dot product to cosine similarity. In our approach, we adopt this normalization as well. See Appendix A for a study on this correlation.

**Per-target Importance.** The formulation above assigns weights to training samples based on their *average* influence over the target set. However, recent work [Xia et al., 2024, Ivison et al., 2025] has shown that selecting training samples based on *per-target* influence can yield better performance; that is, iterating over individual target samples repeatedly and selecting one top-scoring training sample each time. We adopt this approach, noting that influence scores for each target can be computed by running Influence Distillation $|T|$ times—once per target sample.

# 4 Efficient Influence Distillation

In this section, we tackle several challenges regarding the implementation of Influence Distillation.

**Cost of Hessian.** While $\mathbf{Q}(\boldsymbol{\theta})$ can be calculated exactly using Hessian-Vector Products (HVPs), these HVPs require storing the backward graph, which can incur extra memory costs in practice.

**Cost of $\mathbf{G}_S$.** Constructing the matrix $\mathbf{G}_S$ requires computing the gradient of the model with respect to each individual sample in the training set. This process is computationally expensive, as it incurs a cost similar to one full epoch of training on the entire dataset $S$. Furthermore, storing the matrix $\mathbf{G}_S$ requires memory proportional to $|S|$ times the size of the model, which is intractable in practice.

**Regularization Coefficient** The solution to Equation 7 is sensitive to the choice of regularization strength $\lambda$. A key challenge, therefore, is determining how to select $\lambda$ in a practical and effective way.

## 4.1 First-order Influence Distillation

Recall the definition of $f(\boldsymbol{\theta}; \boldsymbol{w})$ from Equation 5, where the second-order term is scaled by the learning rate $\eta$. In Appendix H, we observe that in our settings, $\eta$ is small enough that the second-order term becomes negligible. As a result, computing $\mathbf{Q}$ can be avoided with little to no loss in performance. This first-order approximation aligns with prior work on gradient-based influence estimation, such as the methods proposed by Xia et al. [2024].

## 4.2 Gradient Projection

To reduce the cost of storing the gradients, we take a similar approach to Xia et al. [2024] and project each gradient vector upon calculation into a $k$-dimensional space, where $k \ll d$. As opposed to the mentioned work, which uses random projections sampled from the Rademacher distribution ($\pm 1$ with equal probability), we find that projection using a Randomized Hadamard Transform is faster in practice. For more details on the projections, see Appendix I.

## 4.3 Landmark-Based Gradient Approximation

To circumvent the need to compute gradients for every training sample, we introduce a *landmark-based* approach. At a high level, this method provides an efficient low-rank approximation of the gradient matrix $\mathbf{G}_S$, given by $\hat{\mathbf{G}}_S = \mathbf{C}\mathbf{G}_L$, where $\mathbf{G}_L \in \mathbb{R}^{\ell \times d}$ contains the gradients of $\ell \ll |S|$ selected *landmark* samples. The matrix $\mathbf{C} \in \mathbb{R}^{|S| \times \ell}$ holds the coefficients that express each sample's gradient as a linear combination of the landmark gradients.

Specifically, let $L \subseteq S$ denote a set of $\ell$ landmark samples (e.g., selected at random), and suppose we have access to low-cost per-sample embeddings, represented by $\mathbf{E}_S \in \mathbb{R}^{|S| \times e}$. As before, we assume that all embedding and gradient vectors are normalized. To compute the coefficient matrix $\mathbf{C}$, we minimize the objective $\min_{\mathbf{C} \in \mathbb{R}^{n \times \ell}} ||\mathbf{E}_S - \mathbf{C}\mathbf{E}_L||_2^2$, where $\mathbf{E}_L \in \mathbb{R}^{\ell \times e}$ contains the embeddings of the landmark samples. In words, this procedure approximates each sample's embedding as a linear combination of landmark embeddings. We then estimate the $i$-th row of the gradient matrix, $\mathbf{g}_i$, by $\hat{\mathbf{g}}_i = \mathbf{G}_L^T \mathbf{c}_i$, where $\mathbf{c}_i$ is the $i$-th row of $\mathbf{C}$. This approximation implicitly assumes that the linear relationships learned in the embedding space transfer to the gradient space.

**Theoretical Justification.** Although this approximation is not expected to recover the true gradients with high accuracy, the key intuition is that, as long as it is unbiased, even a weak recovery can yield similar per-sample weights in high-dimensional spaces. Theorem 4.1 demonstrates this property for the specific case of the first-order variant of Influence Distillation.

**Theorem 4.1.** *(Informal version of Theorem D.3 and Corollary D.4 tailored to landmark-based approximation – see Appendix D) Consider the special case of first-order Influence Distillation. Let $\mathbf{g}_i$ and $\hat{\mathbf{g}}_i$ denote the true and the landmark-based approximated gradients for sample $i$, and assume Influence Distillation with $\mathbf{G}_S$ and $\hat{\mathbf{G}}_S$ results in sample weights of $\boldsymbol{w}$ and $\hat{\boldsymbol{w}}$. Further assume:*

- *Unbiased: $\forall i \in \{1, 2, \ldots, n\} : \mathbb{E}[\hat{\mathbf{g}}_i] = \mathbf{g}_i$, i.e., the approximation is unbiased.*

- *Bounded Low-rank MSE: Let $\delta_i^2 = \mathbb{E}[||\hat{\mathbf{g}}_i - \mathbf{g}_i||^2]$, and for some $\Delta^2 \geq 0$: $\frac{1}{n}\sum_{i=1}^{n} \delta_i^2 \leq \Delta^2$.*

*Then* $\mathbb{E}[||\boldsymbol{w} - \hat{\boldsymbol{w}})||^2] \leq \frac{|S|\Delta^2}{\lambda^2 d}$, *with $\lambda$ being the regularization coefficient in Influence Distillation.*

This theorem relates the accuracy of the weights to the low-rank approximation error $\Delta^2$, given the training set size $|S|$, dimension $d$, and regularization parameter $\lambda$. If the approximations are unbiased, in high dimension $d$, it suffices to reasonably control $\Delta^2$ in order to recover the correct weights.

**Integration with Influence Distillation.** Given a low-rank approximation of the gradient matrix $\mathbf{G}_S \approx \hat{\mathbf{G}}_S = \mathbf{C}\mathbf{G}_L$, one can define approximations to objects $\mathbf{p}$ and $\mathbf{Q}$ as below:

$$\hat{\mathbf{p}}(\boldsymbol{\theta}; S, T, L) = \mathbf{C}\mathbf{G}_L(\boldsymbol{\theta})\mathbf{g}_T(\boldsymbol{\theta}) = \mathbf{C} \cdot \mathbf{p}(\boldsymbol{\theta}; L, T), \tag{10}$$

$$\hat{\mathbf{Q}}(\boldsymbol{\theta}; S, T, L) = \frac{1}{|S|}\mathbf{C}\mathbf{G}_L\mathbf{H}_T(\boldsymbol{\theta})\mathbf{G}_L^T\mathbf{C}^T(\boldsymbol{\theta}) = \frac{|L|}{|S|}\mathbf{C} \cdot \mathbf{Q}(\boldsymbol{\theta}; L, T) \cdot \mathbf{C}^T, \tag{11}$$

where $\mathbf{C} = \min_{\mathbf{C}\in\mathbb{R}^{n\times\ell}}||\mathbf{E}_S - \mathbf{C}\mathbf{E}_L||_2^2$ is the coefficient matrix, defined above. As shown in Equations 10 and 11, the landmark-based Influence Distillation computes $\mathbf{p}$ and $\mathbf{Q}$ only for the landmark points, and then propagates them to the remaining samples.

**JVP Embeddings.** We observe that existing embedding methods perform poorly in this setting, exhibiting weak correlation with the true gradients (see Appendix F for a detailed empirical analysis). To address this issue, we introduce *Jacobian-vector Product (JVP) Embeddings*.

Given a sample $x \in S$, we define its JVP embedding as:

$$h_{JVP}(x; \mathcal{N}, \ell, V) = \frac{1}{|V|}\sum_{\mathbf{v}\in V}\frac{\partial\mathcal{N}_\ell(x)}{\partial\boldsymbol{\theta}_\ell} \cdot \mathbf{v} \tag{12}$$

where $\mathcal{N}$ is the model being trained, $\mathcal{N}_\ell(\cdot)$ represents the logits of the next predicted token after processing $x$ through the first $\ell$ layers (or transformer blocks, in case of LLMs), and $\boldsymbol{\theta}_\ell$ are the parameters of these $\ell$ layers. The set $V$ contains random Gaussian vectors matching the shape of $\boldsymbol{\theta}_\ell$, and the term $\frac{\partial\mathcal{N}_\ell(x)}{\partial\boldsymbol{\theta}_\ell}$ is the Jacobian of $\mathcal{N}_\ell(x)$ with respect to $\theta_\ell$. In words, JVP embeddings project the Jacobian of an intermediate model output onto a set of random directions in parameter space.

### 4.4 Tuning the Regularization Coefficient.

Finally, we describe our approach for selecting the regularization coefficient $\lambda$ in Equation 7. As detailed in Appendix G, when $\eta$ is small enough for the second-order term to be negligible and $\lambda = 0$, the solution assigns all the weight to a single sample. As $\lambda$ increases, the solution becomes progressively less sparse, distributing weight across more samples. In the limit $\lambda \to \infty$, the solution becomes fully dense, assigning equal weight to all samples. In practice, given a budget of $k$ samples to select for training, we tune $\lambda$ via binary search to achieve a target sparsity level of $\frac{|S|-k}{|S|}$, thereby ensuring that exactly $k$ samples receive non-zero weight, which we will pick for training.

## 5 Experiments

In this section, we evaluate Influence Distillation across several challenging tasks. We start by detailing the datasets, models, and hyperparameters used in our experiments. Then we present our main results and ablations. Further studies are included in Appendix.

### 5.1 Setting

We largely follow the experimental setup of Ivison et al. [2025] and reuse their code.

**Training Dataset.** We use Tulu V2 [Ivison et al., 2023], a combination of 9 instruction-tuning datasets containing approximately 5.8 million samples. Detailed descriptions of each component dataset are provided in Appendix E. Unless stated otherwise, we randomly sample 200k examples from Tulu V2, and then use sampling methods to pick a subset of 10k samples from this pool.

**Target Datasets.** We evaluate on six target datasets: MMLU [Hendrycks et al., 2021a,b], GSM8k [Cobbe et al., 2021], BBH [Suzgun et al., 2022], TyDIQA Clark et al. [2020], Codex [Chen et al.,

Table 1: Accuracy (± standard deviation) and estimated embedding and selection cost (Embd+Sel, in TF, TeraFLOPs) of various methods across tasks and models. For each model–dataset pair, 10k training samples are selected from a pool of size $200k$ from the Tulu V2 dataset [Ivison et al., 2023]. We additionally report average improvement over the Uniform baseline (Avg. $\Delta$ w/ Uniform). Top performing selection methods, as well as Full training numbers are in bold.

| Model | Method | MMLU | GSM8k | BBH | TyDIQA | CODEX | SQuAD | Avg. $\Delta$ w/ Uniform | Embd+Sel Cost |
|---|---|---|---|---|---|---|---|---|---|
| Llama2-7B | Uniform | $45.6 \pm 0.43$ | $17.5 \pm 1.08$ | $41.8 \pm 0.20$ | $51.6 \pm 0.38$ | $27.0 \pm 0.60$ | $80.8 \pm 1.05$ | 0.00 | 0 |
| | Mid-PPL | $45.6 \pm 0.86$ | $15.0 \pm 0.54$ | $40.9 \pm 0.23$ | $52.1 \pm 0.44$ | $26.1 \pm 1.56$ | $80.7 \pm 0.73$ | -0.65 | 2800 TF |
| | RDS+ | $46.3 \pm 0.33$ | $20.2 \pm 2.77$ | $42.7 \pm 0.61$ | $50.5 \pm 0.84$ | $\mathbf{30.4 \pm 0.96}$ | $\mathbf{85.3 \pm 0.22}$ | +1.85 | 2800 TF |
| | InfDist | $\mathbf{48.3 \pm 0.21}$ | $\mathbf{20.3 \pm 1.65}$ | $\mathbf{43.2 \pm 0.67}$ | $\mathbf{53.6 \pm 0.34}$ | $29.5 \pm 3.14$ | $83.2 \pm 1.02$ | **+2.30** | **872 TF** |
| | **Full** | $\mathbf{48.8 \pm 0.08}$ | $\mathbf{21.2 \pm 0.85}$ | $\mathbf{43.9 \pm 0.24}$ | $\mathbf{51.3 \pm 0.18}$ | $\mathbf{29.3 \pm 3.72}$ | $\mathbf{83.6 \pm 0.30}$ | **+2.30** | − |
| Llama3.2-3B | Uniform | $53.9 \pm 0.52$ | $34.6 \pm 1.22$ | $48.9 \pm 0.67$ | $63.1 \pm 0.36$ | $56.1 \pm 1.35$ | $80.4 \pm 0.51$ | 0.00 | 0 |
| | Mid-PPL | $\mathbf{54.0 \pm 0.27}$ | $29.5 \pm 0.12$ | $48.3 \pm 0.44$ | $\mathbf{65.9 \pm 0.66}$ | $55.9 \pm 4.40$ | $80.9 \pm 0.18$ | -0.42 | 1200 TF |
| | RDS+ | $53.1 \pm 0.58$ | $\mathbf{38.4 \pm 0.58}$ | $\mathbf{49.6 \pm 0.45}$ | $61.0 \pm 0.35$ | $\mathbf{60.6 \pm 1.77}$ | $\mathbf{84.2 \pm 0.47}$ | +1.65 | 1200 TF |
| | InfDist | $\mathbf{54.0 \pm 0.94}$ | $35.7 \pm 1.28$ | $48.6 \pm 0.27$ | $64.6 \pm 1.29$ | $55.4 \pm 1.10$ | $83.3 \pm 1.97$ | +0.77 | **417 TF** |
| | **Full** | $52.9 \pm 0.87$ | $37.0 \pm 0.33$ | $48.9 \pm 0.14$ | $62.5 \pm 1.50$ | $57.7 \pm 2.81$ | $83.9 \pm 1.15$ | +0.98 | − |
| Qwen2.5-1.5B | Uniform | $58.8 \pm 0.11$ | $\mathbf{63.3 \pm 1.67}$ | $44.1 \pm 0.38$ | $55.0 \pm 0.32$ | $70.5 \pm 2.06$ | $16.5 \pm 4.65$ | 0.00 | 0 |
| | Mid-PPL | $58.9 \pm 0.17$ | $63.2 \pm 0.65$ | $\mathbf{44.3 \pm 0.31}$ | $54.3 \pm 0.29$ | $70.6 \pm 1.61$ | $22.3 \pm 3.39$ | +0.90 | 600 TF |
| | RDS+ | $58.3 \pm 0.07$ | $60.1 \pm 0.13$ | $44.2 \pm 0.24$ | $53.0 \pm 0.38$ | $\mathbf{72.3 \pm 0.00}$ | $46.0 \pm 3.12$ | +4.28 | 600 TF |
| | InfDist | $\mathbf{59.4 \pm 0.12}$ | $62.0 \pm 0.46$ | $44.1 \pm 0.35$ | $\mathbf{57.6 \pm 0.32}$ | $69.8 \pm 1.15$ | $54.4 \pm 13.13$ | **+6.52** | **208 TF** |
| | **Full** | $\mathbf{59.4 \pm 0.13}$ | $60.3 \pm 0.38$ | $44.0 \pm 0.22$ | $50.3 \pm 1.21$ | $73.0 \pm 1.79$ | $63.2 \pm 6.05$ | **+7.00** | − |
| Qwen2.5-3B | Uniform | $63.7 \pm 0.27$ | $68.7 \pm 1.87$ | $54.9 \pm 0.24$ | $65.6 \pm 0.16$ | $83.1 \pm 1.35$ | $84.5 \pm 0.54$ | 0.00 | 0 |
| | Mid-PPL | $63.7 \pm 0.18$ | $\mathbf{70.8 \pm 0.84}$ | $\mathbf{55.1 \pm 0.21}$ | $65.3 \pm 0.41$ | $\mathbf{83.3 \pm 3.14}$ | $79.9 \pm 2.61$ | -0.40 | 1200 TF |
| | RDS+ | $63.6 \pm 0.19$ | $67.4 \pm 0.68$ | $54.0 \pm 0.63$ | $65.1 \pm 0.33$ | $82.4 \pm 1.46$ | $\mathbf{86.3 \pm 0.33}$ | -0.28 | 1200 TF |
| | InfDist | $\mathbf{64.6 \pm 0.19}$ | $67.8 \pm 0.60$ | $53.9 \pm 0.36$ | $\mathbf{66.9 \pm 0.23}$ | $82.4 \pm 0.55$ | $86.0 \pm 0.22$ | **+0.18** | **340 TF** |
| | **Full** | $\mathbf{63.8 \pm 0.06}$ | $\mathbf{71.0 \pm 1.78}$ | $53.8 \pm 0.32$ | $64.5 \pm 0.42$ | $82.0 \pm 1.41$ | $85.4 \pm 0.42$ | 0.00 | − |

2021], and SQuAD [Rajpurkar et al., 2016]. For each, we assume access to 8–500 examples from their train, dev, or eval splits [3]. Details are in Appendix E.

**Model.** We mainly consider fine-tuning the LlaMA-2 7B model [Touvron et al., 2023], consistent with the Tulu V2 paper [Ivison et al., 2023] and the experiments of Ivison et al. [2025]. We also experiment with Llama-3.2 3B [Grattafiori et al., 2024] and Qwen 2.5 1.5/3B [Team, 2024].

**Baselines.** We consider four main baselines: (1) Random *Uniform* selection, which picks samples uniformly at random, (2) The state-of-the-art *RDS+* [Ivison et al., 2025] embedding-based method, where the embeddings are computed by a position-weighted mean pool of the last hidden layer states, (3) *Mid-PPL* [Yin and Rush, 2024], where samples are sorted by their perplexity, and the middle ones are selected, and (4) *Full*, where we do not perform any sampling and train on the full dataset. Additionally, we also include a comparison with LESS [Xia et al., 2024], which is similar to first-order Influence Distillation but requires exact gradients for each sample.

**Hyperparameters.** We use the AdamW optimizer with a learning rate of $2 \times 10^{-5}$ and a linear schedule for 2 epochs. The sequence length is fixed at 2048, and we use a microbatch size of 1 with gradient accumulation over 128 steps. All experiments are conducted on a single H100 GPU, and each are repeated with 3 random seeds, including the selection of 200k samples from Tulu V2.

By default, we use first-order Influence Distillation with 4096 landmarks. We select the landmarks uniformly at random, as we find this performs comparably to more complex methods such as leverage score sampling (see Appendix K.4). Linear coefficients are computed via Kernel Ridge Regression (KRR) with an RBF kernel and dampening of 0.01. JVP embeddings are obtained from the first four transformer blocks using two random vectors ($\ell = 4$, $|V| = 2$), following a brief warm-up on 10k random samples. The model is then reset and trained on the selected subset. This warm-up is needed to stabilize gradients (see Appendix A). Gradients are projected to 131072 dimensions via Hadamard projections; we use the largest dimension that fits in GPU memory, as projection cost does not depend on the dimension (Appendix I). After selection, we do not incorporate the sample weights during training, as experiments in Appendix J suggest this does not improve performance.

## 5.2 Results

**Main Experiments.** Table 1 summarizes our main experimental results. In each case, a subset of size 10k is selected from a pool of 200k Tulu V2 [Ivison et al., 2023] samples. On average, Influence Distillation achieves higher performance compared to other more expensive selection baselines in three out of four models and remains competitive in the fourth, while enabling 2.9–3.5$\times$ faster

---

[3]Following Ivison et al. [2025], except we omit AlpacaEval [Li et al., 2023b] as it requires paid API access.

selection. Notably, for two models, it matches or surpasses training on the full dataset. These results clearly showcase the effectiveness and efficiency of Influence Distillation.

**Selection Runtime Estimation.** The table also reports the approximate FLOPs required for sample selection. Following the estimation from Kaplan et al. [2020], each forward pass is assumed to cost $2d$ FLOPs and each backward pass $4d$ FLOPs, where $d$ is the number of model parameters. Mid-PPL and RDS+ require one forward pass per sample. Influence Distillation requires computing a JVP embedding for each sample, along with full forward and backward passes for the 4096 selected landmarks. We estimate the cost of a JVP as $2\times$ that of a partial forward pass over the same number of blocks, following Cobb et al. [2024].

**Pareto Superiority.** We repeat the above experiments on the Llama2-7B model using pool sizes 50k, 100k, 150k, and 200k. For each pool size, we select 2048, 4096, 6144, and 8192 landmarks, respectively, maintaining a fixed pool-to-landmark ratio. As shown in Figure 1, all points corresponding to Influence Distillation lie on the Pareto frontier, matching or surpassing the performance of RDS+ while requiring lower overall cost of embedding, sampling, and training.

**Effect of Number of Landmarks.** To evaluate how the number of landmarks impacts performance, we repeat the experiments on Llama2-7B and report average improvement over the Uniform baseline across the six tasks in Figure 3 (Left). As shown, Influence Distillation improves with more landmarks, surpassing RDS+ beyond 2048 landmarks.

**Effect of Pool Size and Number of Selected Samples.** Figure 3 (Right) presents a heatmap of MMLU accuracy on Llama2-7B across different combinations of pool size (up to 200k) and number of selected samples. For each pool size, we use the same number of landmarks as the Pareto experiment. As expected, accuracy improves with larger pools and more selected samples, highlighting the scalability and robustness of Influence Distillation.

**Comparison with LESS.** We ran the 10k/200k selection experiment on Llama2-7B [Touvron et al., 2023] using the LESS [Xia et al., 2024] method (with one seed due to its high computational cost). In this setup, gradients are computed after a 10k-step warmup, projected down to $8192$ dimensions [Park et al., 2023]. The top 10k examples are then selected based on similarity with the target gradients, following the original LESS recipe. Table 2 presents a comparison between Uniform, Influence Distillation with JVP embeddings, LESS, and Full.

Table 2: Comparison with LESS [Xia et al., 2024]

| Model | Method | MMLU | GSM8k | BBH | TyDIQA | CODEX | SQuAD | Avg. $\Delta$ w/ Uniform | Embd+Sel Cost |
|---|---|---|---|---|---|---|---|---|---|
| Llama2-7B | Uniform | $45.6 \pm 0.43$ | $17.5 \pm 1.08$ | $41.8 \pm 0.20$ | $51.6 \pm 0.38$ | $27.0 \pm 0.60$ | $80.8 \pm 1.05$ | 0.00 | 0 |
| | InfDist | $\mathbf{48.3 \pm 0.21}$ | $20.3 \pm 1.65$ | $43.2 \pm 0.67$ | $53.6 \pm 0.34$ | $\mathbf{29.5 \pm 3.14}$ | $83.2 \pm 1.02$ | +2.30 | **872 TF** |
| | LESS | 48.2 | **22.7** | **43.4** | **55.8** | 29.1 | **86.0** | +3.08 | 8400 TF |
| | Full | $\mathbf{48.8 \pm 0.08}$ | $\mathbf{21.2 \pm 0.85}$ | $\mathbf{43.9 \pm 0.24}$ | $51.3 \pm 0.18$ | $\mathbf{29.3 \pm 3.72}$ | $\mathbf{83.6 \pm 0.30}$ | +2.30 | — |

While LESS achieves higher average accuracy, it is roughly ten times more computationally expensive than our method, requiring about $8400$ TeraFLOPs (TFs) compared to our $872$ TFs for embedding and selection. Its computational cost is prohibitively high in realistic settings, as it requires computing and projecting gradients for every sample in the pool, followed by fine-tuning on the selected subset. Even when disregarding the significant cost of gradient projection, the overall computation still exceeds that of training on the full dataset for one epoch. A similar observation was made by Ivison et al. [2025], who did not compare with LESS in large-scale experiments due to its high computational cost. This limitation underscores that LESS is not practically comparable to Influence Distillation, which achieves efficiency through landmark-based approximation.

**Additional Studies.** We present a comprehensive set of additional empirical studies in Appendix K. These studies investigate topics such as weight transferability across models and tasks, the impact of the target set size, and various landmark selection methods, among others.

# 6 Limitations and Future Work

Below, we outline three main limitations of our work, along with corresponding directions for future research.

**No Target Distribution.** While we demonstrate that Influence Distillation is highly effective for targeted instruction tuning across a range of models and tasks, it does not directly extend to general

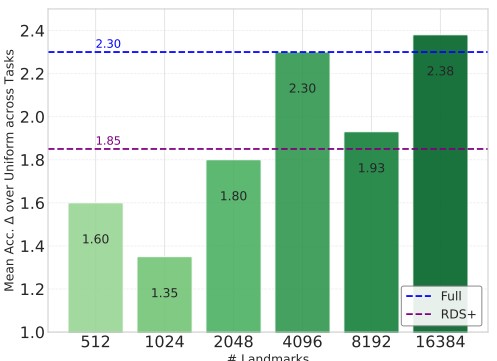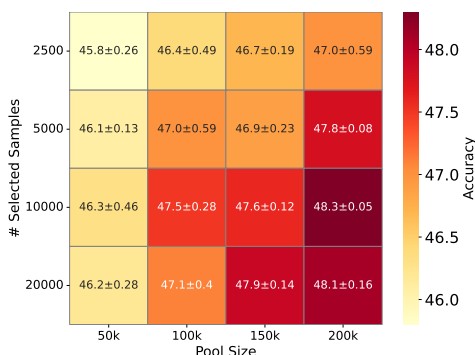

Figure 3: (Left) Effect of the number of landmarks on the performance of Influence Distillation across six tasks using Llama2-7B. (Right) MMLU accuracy of Influence Distillation on Llama2-7B across different pool sizes and number of selected samples.

data selection scenarios where no target dataset is available. In such cases, one could define the target distribution as a small set of high-quality examples or a representative subset of the training corpus. Investigating how to construct and utilize such proxy targets is an important direction for future work.

**Pre-training.** Extending Influence Distillation to the pre-training setting presents unique challenges. In particular, the considerably longer duration of pre-training implies that gradients may shift substantially over time, likely making a single static selection insufficient. This suggests the need for a multi-phase selection strategy, such as periodic re-sampling. We leave the exploration of such dynamic approaches to future work.

**Warm-up Cost.** We exclude the cost of the warm-up phase from our runtime measurements for two reasons: (1) as the training pool grows, the cost of a brief warm-up on a small random subset becomes negligible compared to embedding the full dataset; and (2) the warm-up can be shortened (as shown in Appendix A) or compressed—for example, via Low-Rank Adaptation [Hu et al., 2022], as in Xia et al. [2024]. We leave a rigorous investigation of warm-up optimization to future work.

## Acknowledgments

We would like to thank ISTA SciComp for providing access to their GPU resources. We also wish to thank Baharan Mirzasoleiman for their insightful feedback and helpful discussions.

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

# A Gradient Analysis During Training

In this section, we analyze the behavior of gradients throughout training. We fine-tune a LLaMA2-7B model [Touvron et al., 2023] on 10000 randomly selected samples from Tulu V2 [Ivison et al., 2023] for 2 epochs, saving model checkpoints every 10 steps.

For each checkpoint—including the initial and final models—we compute the gradients of 1000 held-out samples from Tulu V2, as well as samples from the target dataset BBH [Suzgun et al., 2022], and project them into an 8192-dimensional space using random Rademacher matrices, following the efficient GPU implementation of Park et al. [2023], also adopted in Xia et al. [2024]. For each dataset, we compute the average gradient cosine similarity across checkpoints. As shown in Figure 4, while the gradient directions can change substantially in the early steps, they stabilize quickly during training. This observation justifies the use of a short warm-up phase as both necessary and sufficient. Similar plots for GSM8k [Cobbe et al., 2021] and SQuAD [Rajpurkar et al., 2016] are provided later in the Appendix (Figure 9).

Additionally, for each dataset and checkpoint, we measure the Pearson product-moment correlation between gradient norms and the number of label tokens per sample. As shown in Figure 5, we observe a consistent negative correlation, which supports our decision to normalize gradients prior to distillation.

# B Linear Model Study

In this section, we show that a regularization term can be effective in robustifying Objective 6 to small changes in the model parameters $\boldsymbol{\theta}$, when the model is linear and the loss is quadratic.

For a fixed $\epsilon > 0$, define a new objective as below:

$$\boldsymbol{w}^* = \arg\min_{\boldsymbol{w}} \max_{||\boldsymbol{\delta}|| \leq \epsilon} f(\boldsymbol{w}; \boldsymbol{\theta} + \boldsymbol{\delta}), \tag{13}$$

minimizing the maximum value of $f$ around a point $\boldsymbol{\theta}$ in a neighborhood of radius $\epsilon$. This ensures the weights are stable as long as $\boldsymbol{\theta}$ is in this neighborhood. To solve for $w$, we employ Lemma B.1 below:

**Lemma B.1.** *Assume* $\mathcal{L}(\boldsymbol{\theta}; D, \boldsymbol{w}) = \sum_{i=1}^{|D|} \boldsymbol{w}_i(\langle \mathbf{x}_i^D, \boldsymbol{\theta} \rangle - y_i^D)^2$, *where* $D = \{(\mathbf{x}_1^D, y_1^D), (\mathbf{x}_2^D, y_2^D), ..., (\mathbf{x}_{|D|}^D, y_{|D|}^D)\}$ *is a dataset. For datasets $S$ and $T$, let* $\mathbf{H}_T = \nabla_{\boldsymbol{\theta}}^2 \mathcal{L}(\boldsymbol{\theta}; T, \mathbb{1})$, $\mathbf{g}_T = \nabla_{\boldsymbol{\theta}} \mathcal{L}(\boldsymbol{\theta}; T, \mathbb{1})$, $\mathbf{H}_{\boldsymbol{w}} = \nabla_{\boldsymbol{\theta}}^2 \mathcal{L}(\boldsymbol{\theta}; S, \boldsymbol{w})$, *and* $\mathbf{g}_{\boldsymbol{w}} = \nabla_{\boldsymbol{\theta}} \mathcal{L}(\boldsymbol{\theta}; S, \boldsymbol{w})$. *Define* $\mathbf{a}_{\boldsymbol{w}}$ *and* $\mathbf{B}_{\boldsymbol{w}}$ *as below:*

$$\mathbf{a}_{\boldsymbol{w}} = -\mathbf{H}_{\boldsymbol{w}}\mathbf{g}_T - \mathbf{H}_T\mathbf{g}_{\boldsymbol{w}} + \eta \mathbf{H}_{\boldsymbol{w}}\mathbf{H}_T\mathbf{g}_{\boldsymbol{w}} \tag{14}$$

$$\mathbf{B}_{\boldsymbol{w}} = -\mathbf{H}_T\mathbf{H}_{\boldsymbol{w}} + \frac{\eta}{2}\mathbf{H}_{\boldsymbol{w}}\mathbf{H}_T\mathbf{H}_{\boldsymbol{w}} \tag{15}$$

*In the setting above (linear model with quadratic loss), the function $f$ has the property that $\forall\, \boldsymbol{\theta}, \boldsymbol{\delta} \in \mathbb{R}^d, \boldsymbol{w} \in \mathbb{R}^n$:*

$$f(\boldsymbol{w}; \boldsymbol{\theta} + \boldsymbol{\delta}) = f(\boldsymbol{w}; \boldsymbol{\theta}) + \mathbf{a}_{\boldsymbol{w}}^T \boldsymbol{\delta} + \boldsymbol{\delta}^T \mathbf{B}_{\boldsymbol{w}} \boldsymbol{\delta}. \tag{16}$$

*Proof.* First notice that, for simplicity, the loss here is defined as the sum (as opposed to the average) of per-sample losses, which drops the $\frac{1}{|S|}$ terms in the loss, gradient, Hessian, and $\mathbf{Q}$ objects. Recalling the definition of $f$ from 5, we can write $f(\boldsymbol{w}; \boldsymbol{\theta} + \boldsymbol{\delta}) = -\mathbf{p}(\boldsymbol{\theta} + \boldsymbol{\delta})^T \boldsymbol{w} + \frac{\eta}{2} \boldsymbol{w}^T \mathbf{Q}(\boldsymbol{\theta} + \boldsymbol{\delta}) \boldsymbol{w}$. Since the loss is quadratic in $\boldsymbol{\theta}$, the Hessian is independent of $\boldsymbol{\theta}$, and the derivatives above the second order are zero. Hence, defining $\mathbf{g}_i^D(\boldsymbol{\theta})$ and $\mathbf{H}_i^D$ as the gradient and Hessian of the sample $i$ in $D$, we can write:

$$\mathbf{g}_i^D(\boldsymbol{\theta} + \boldsymbol{\delta}) = \mathbf{g}_i^D(\boldsymbol{\theta}) + \mathbf{H}_i^D \boldsymbol{\delta} \tag{17}$$

for any $\boldsymbol{\delta}$ with the same dimension as $\boldsymbol{\theta}$. Setting $D = T$ and summing across samples, we can write:

$$\mathbf{g}_T(\boldsymbol{\theta} + \boldsymbol{\delta}) = \mathbf{g}_T(\boldsymbol{\theta}) + \mathbf{H}_T \boldsymbol{\delta} \tag{18}$$

Additionally, setting $D = S$ and taking a weighted sum we can write:

$$\mathbf{G}_S(\boldsymbol{\theta} + \boldsymbol{\delta})\boldsymbol{w} = \mathbf{G}_S(\boldsymbol{\theta})\boldsymbol{w} + \mathbf{H}_{\boldsymbol{w}} \boldsymbol{\delta} \tag{19}$$

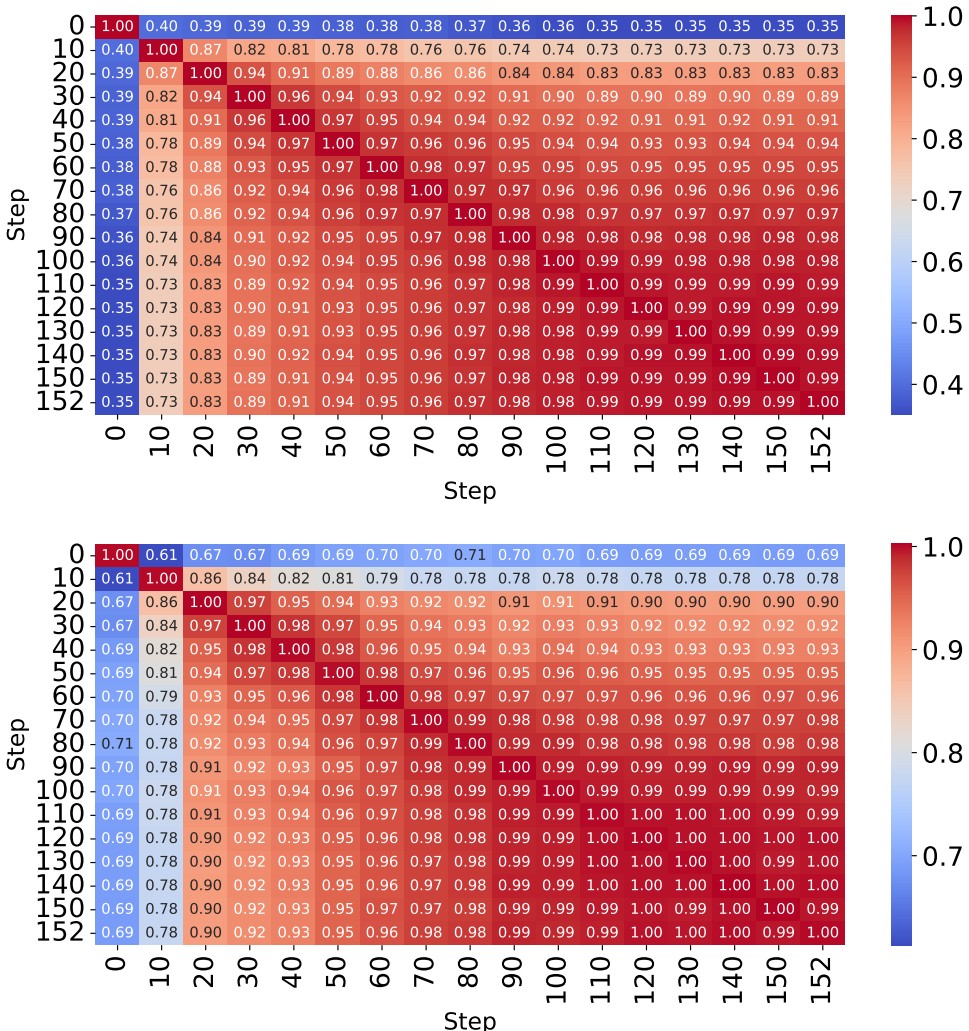

Figure 4: Average gradient cosine similarity on unseen samples from Tulu V2 (top) and BBH (bottom) across checkpoints.

Next, we see that,

$$\mathbf{p}(\boldsymbol{\theta} + \boldsymbol{\delta})^T \boldsymbol{w} = \mathbf{g}_T(\boldsymbol{\theta} + \boldsymbol{\delta})\mathbf{G}_S(\boldsymbol{\theta} + \boldsymbol{\delta})\boldsymbol{w}$$
$$= (\mathbf{g}_T(\boldsymbol{\theta})^T + \boldsymbol{\delta}^T \mathbf{H}_T)(\mathbf{G}_S(\boldsymbol{\theta})\boldsymbol{w} + \mathbf{H}_{\boldsymbol{w}}\boldsymbol{\delta})$$
$$= \mathbf{p}(\boldsymbol{\theta})^T \boldsymbol{w} + (\mathbf{g}_T(\boldsymbol{\theta})^T \mathbf{H}_{\boldsymbol{w}} + \mathbf{g}_{\boldsymbol{w}}(\boldsymbol{\theta})^T \mathbf{H}_T)\boldsymbol{\delta} + \boldsymbol{\delta}^T \mathbf{H}_T \mathbf{H}_{\boldsymbol{w}}\boldsymbol{\delta} \qquad (20)$$

And,

$$\boldsymbol{w}^T \mathbf{Q}(\boldsymbol{\theta} + \boldsymbol{\delta})^T \boldsymbol{w} = \boldsymbol{w}^T \mathbf{G}_S(\boldsymbol{\theta} + \boldsymbol{\delta})^T \mathbf{H}_T \mathbf{G}_S(\boldsymbol{\theta} + \boldsymbol{\delta})\boldsymbol{w}$$
$$= (\boldsymbol{w}^T \mathbf{G}_S(\boldsymbol{\theta})^T + \boldsymbol{\delta}^T \mathbf{H}_{\boldsymbol{w}})\mathbf{H}_T(\mathbf{G}_S(\boldsymbol{\theta})\boldsymbol{w} + \mathbf{H}_{\boldsymbol{w}}\boldsymbol{\delta})$$
$$= \boldsymbol{w}^T \mathbf{Q}(\boldsymbol{\theta})\boldsymbol{w} + 2\mathbf{g}_{\boldsymbol{w}}(\boldsymbol{\theta})^T \mathbf{H}_T \mathbf{H}_{\boldsymbol{w}}\boldsymbol{\delta} + \boldsymbol{\delta}^T \mathbf{H}_{\boldsymbol{w}} \mathbf{H}_T \mathbf{H}_{\boldsymbol{w}}\boldsymbol{\delta} \qquad (21)$$

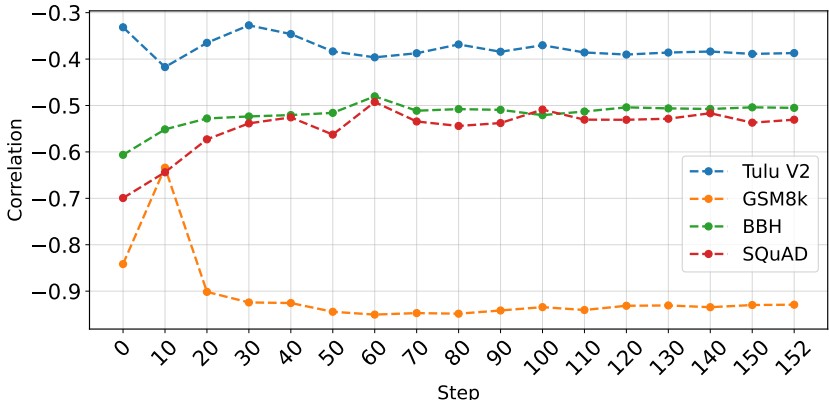

Figure 5: Correlation between gradient norm and number of label tokens, across checkpoints on four datasets.

Putting them together:

$$
\begin{aligned}
f(\boldsymbol{w}; \boldsymbol{\theta} + \boldsymbol{\delta}) &= -\mathbf{p}(\boldsymbol{\theta} + \boldsymbol{\delta})^T \boldsymbol{w} + \frac{\eta}{2} \boldsymbol{w}^T \mathbf{Q}(\boldsymbol{\theta} + \boldsymbol{\delta}) \boldsymbol{w} \\
&= f(\boldsymbol{w}; \boldsymbol{\theta}) - ((\mathbf{g}_T(\boldsymbol{\theta})^T \mathbf{H}_{\boldsymbol{w}} + \mathbf{g}_{\boldsymbol{w}}(\boldsymbol{\theta})^T \mathbf{H}_T) \boldsymbol{\delta} + \boldsymbol{\delta}^T \mathbf{H}_T \mathbf{H}_{\boldsymbol{w}} \boldsymbol{\delta}) \\
&\quad + \frac{\eta}{2} (2 \mathbf{g}_{\boldsymbol{w}}(\boldsymbol{\theta})^T \mathbf{H}_T \mathbf{H}_{\boldsymbol{w}} \boldsymbol{\delta} + \boldsymbol{\delta}^T \mathbf{H}_{\boldsymbol{w}} \mathbf{H}_T \mathbf{H}_{\boldsymbol{w}} \boldsymbol{\delta}) \\
&= f(\boldsymbol{w}; \boldsymbol{\theta}) + (-\mathbf{g}_T(\boldsymbol{\theta})^T \mathbf{H}_{\boldsymbol{w}} - \mathbf{g}_{\boldsymbol{w}}(\boldsymbol{\theta})^T \mathbf{H}_T + \eta \mathbf{g}_{\boldsymbol{w}}(\boldsymbol{\theta})^T \mathbf{H}_T \mathbf{H}_{\boldsymbol{w}}) \boldsymbol{\delta} \\
&\quad + \boldsymbol{\delta}^T (-\mathbf{H}_T \mathbf{H}_{\boldsymbol{w}} + \frac{\eta}{2} \mathbf{H}_{\boldsymbol{w}} \mathbf{H}_T \mathbf{H}_{\boldsymbol{w}}) \boldsymbol{\delta} \\
&= f(\boldsymbol{w}; \boldsymbol{\theta}) + \mathbf{a}_{\boldsymbol{w}}^T \boldsymbol{\delta} + \boldsymbol{\delta}^T \mathbf{B}_{\boldsymbol{w}} \boldsymbol{\delta}
\end{aligned}
$$

which concludes the proof. □

Substituting the result of the B.1 into Objective 13, we can write

$$
\begin{aligned}
\boldsymbol{w}^* &= \arg\min_{\boldsymbol{w}} \max_{||\boldsymbol{\delta}|| \leq \epsilon} \left[ f(\boldsymbol{w}; \boldsymbol{\theta}) + \mathbf{a}_{\boldsymbol{w}}^T \boldsymbol{\delta} + \boldsymbol{\delta}^T \mathbf{B}_{\boldsymbol{w}} \boldsymbol{\delta} \right] \\
&= \arg\min_{\boldsymbol{w}} \left[ f(\boldsymbol{w}; \boldsymbol{\theta}) + \max_{||\boldsymbol{\delta}|| \leq \epsilon} (\mathbf{a}_{\boldsymbol{w}}^T \boldsymbol{\delta} + \boldsymbol{\delta}^T \mathbf{B}_{\boldsymbol{w}} \boldsymbol{\delta}) \right]
\end{aligned}
\tag{22}
$$

We maximize $r(\boldsymbol{\delta}) = \mathbf{a}_{\boldsymbol{w}}^T \boldsymbol{\delta} + \boldsymbol{\delta}^T \mathbf{B}_{\boldsymbol{w}} \boldsymbol{\delta}$ in the sphere with radius $\epsilon$ approximately by taking a single step of size $\epsilon$ in the gradient direction, i.e., $\boldsymbol{\delta}^* \approx \epsilon \cdot \frac{r'(\mathbf{0})}{||r'(\mathbf{0})||}$. This approximation is standard in the sharpness-aware optimization literature [Foret et al., 2020, Peste et al., 2022], which addresses a similar min-max objective to search for flat minima. Note that $r'(\boldsymbol{\delta}) = \mathbf{a}_{\boldsymbol{w}} + (\mathbf{B}_{\boldsymbol{w}} + \mathbf{B}_{\boldsymbol{w}}^T) \boldsymbol{\delta}$, hence $r'(\mathbf{0}) = \mathbf{a}_{\boldsymbol{w}}$ and

$$
\max_{||\boldsymbol{\delta}|| \leq \epsilon} (\mathbf{a}_{\boldsymbol{w}}^T \boldsymbol{\delta} + \boldsymbol{\delta}^T \mathbf{B}_{\boldsymbol{w}} \boldsymbol{\delta}) \approx \epsilon \cdot ||\mathbf{a}_{\boldsymbol{w}}|| + \epsilon^2 \cdot \frac{\mathbf{a}_{\boldsymbol{w}}^T \mathbf{B}_{\boldsymbol{w}} \mathbf{a}_{\boldsymbol{w}}}{||\mathbf{a}_{\boldsymbol{w}}||^2}.
\tag{23}
$$

Substituting into Equation 22, we get the following objective:

$$
\boldsymbol{w}^* \approx \arg\min_{\boldsymbol{w}} \left[ f(\boldsymbol{w}; \boldsymbol{\theta}) + \epsilon \cdot ||\mathbf{a}_{\boldsymbol{w}}|| + \epsilon^2 \cdot \frac{\mathbf{a}_{\boldsymbol{w}}^T \mathbf{B}_{\boldsymbol{w}} \mathbf{a}_{\boldsymbol{w}}}{||\mathbf{a}_{\boldsymbol{w}}||^2} \right].
\tag{24}
$$

This suggests that the robustness of the weights can be controlled via the hyperparameter $\epsilon$, which determines the strength of the regularization.

We apply this regularization to the running example introduced in Section 3.2. As shown in Figure 6, using the tuned value $\epsilon = 10^{-4}$ yields better performance than the default weights. However, due to the high computational cost of this regularization term, we use standard L2 regularization for general non-linear models.

## C  Adam Optimizer

Here we derive Equations 8 and 9, which adapt the vector $\mathbf{p}$ and the matrix $\mathbf{Q}$ to the case of the Adam optimizer. Assume that after a warm-up phase, the first- and second-moment estimates of Adam are $\mathbf{m}$ and $\mathbf{v}$, respectively. For a new gradient $\mathbf{g}$, the Adam update rule can be written as:

$$\Delta\boldsymbol{\theta} = -\eta \cdot \frac{\mathbf{m}'}{\sqrt{\mathbf{v}'} + \epsilon} \tag{25}$$

where $\eta$ is the learning rate, and $\mathbf{m}' = \frac{\beta_1\mathbf{m} + (1-\beta_1)\mathbf{g}}{1-\beta_1^s}$ and $\mathbf{v}' = \frac{\beta_2\mathbf{v} + (1-\beta_2)\mathbf{g}^2}{1-\beta_2^s}$ are the updated moment estimates, with $(\beta_1, \beta_2)$ being the Adam beta values for first- and second-order estimate updates, and $s$ being the number steps the optimizer has already been trained for.

For a single update, we note that $\beta_2\mathbf{v} + (1-\beta_2)\mathbf{g}^2 \approx \mathbf{v}$. That is because (1) the value $\beta_2$ is typically very close to 1, e.g., 0.995 or 0.999, and (2) due to the warm-up, $\mathbf{v}$ is stabilized and is not expected to change much. This allows us to ignore the dependence of $\mathbf{v}'$ on $\mathbf{g}$, i.e., $\mathbf{v}' \approx \frac{\mathbf{v}}{1-\beta_2^s}$ simplifying the computations.

Enabled by this, we revisit the Taylor expansion in Equation 2:

$$\boldsymbol{w}^* = \arg\min_{\boldsymbol{w}} \ \mathcal{L}(\mathcal{M}^{\text{Adam}}(\boldsymbol{\theta}; S, \boldsymbol{w}); T, \mathbb{1})$$

$$= \arg\min_{\boldsymbol{w}} \ \mathcal{L}(\boldsymbol{\theta} - \frac{\eta}{|S|} \frac{\frac{\beta_1\mathbf{m} + (1-\beta_1)\mathbf{G}_S(\boldsymbol{\theta})\boldsymbol{w}}{1-\beta_1^s}}{\sqrt{\frac{\mathbf{v}}{1-\beta_2^s}} + \epsilon})$$

$$= \arg\min_{\boldsymbol{w}} \ \mathcal{L}(\boldsymbol{\theta} - \frac{\eta}{|S|}[\frac{\beta_1\mathbf{m}}{(1-\beta_1^s)(\sqrt{\frac{\mathbf{v}}{1-\beta_2^s}} + \epsilon)} + \frac{(1-\beta_1)\mathbf{G}_S(\boldsymbol{\theta})\boldsymbol{w}}{(1-\beta_1^s)(\sqrt{\frac{\mathbf{v}}{1-\beta_2^s}} + \epsilon)}]) \tag{26}$$

Let $\mathbf{a} = \frac{(1-\beta_1)}{(1-\beta_1^s)(\sqrt{\frac{\mathbf{v}}{1-\beta_2^s}}+\epsilon)}$ and $\mathbf{b} = \frac{\beta_1\mathbf{m}}{(1-\beta_1^s)(\sqrt{\frac{\mathbf{v}}{1-\beta_2^s}}+\epsilon)}$. Construct $\mathbf{G}_S^{\text{Adam}}(\boldsymbol{\theta})$ by element-wise multiplying each column of $\mathbf{G}_S(\boldsymbol{\theta})$ by $\mathbf{a}$. We can now continue Equation 26 by:

$$\boldsymbol{w}^* = \arg\min_{\boldsymbol{w}} \ \mathcal{L}(\boldsymbol{\theta} - \frac{\eta}{|S|}(\mathbf{b} + \mathbf{G}_S^{\text{Adam}}(\boldsymbol{\theta})\boldsymbol{w})$$

$$\approx \arg\min_{\boldsymbol{w}} \ [\mathcal{L}(\boldsymbol{\theta}; T, \mathbb{1}) - \frac{\eta}{|S|}\mathbf{g}_T^T(\boldsymbol{\theta})(\mathbf{b} + \mathbf{G}_S^{\text{Adam}}(\boldsymbol{\theta})\boldsymbol{w}) +$$

$$\frac{\eta^2}{2\,|S|^2}(\mathbf{b}^T + \boldsymbol{w}^T\mathbf{G}_S^{\text{Adam}}(\boldsymbol{\theta})^T)\mathbf{H}_T(\boldsymbol{\theta})(\mathbf{b} + \mathbf{G}_S^{\text{Adam}}(\boldsymbol{\theta})\boldsymbol{w})]$$

$$= \arg\min_{\boldsymbol{w}} \ -(\mathbf{g}_T^T(\boldsymbol{\theta}) - \frac{\eta}{|S|}\mathbf{b}^T\mathbf{H}_T(\boldsymbol{\theta}))\mathbf{G}_S^{\text{Adam}}(\boldsymbol{\theta})\boldsymbol{w}$$

$$+ \frac{\eta}{2\,|S|}\boldsymbol{w}^T\mathbf{G}_S^{\text{Adam}}(\boldsymbol{\theta})^T\mathbf{H}_T(\boldsymbol{\theta})\mathbf{G}_S^{\text{Adam}}(\boldsymbol{\theta})\boldsymbol{w}$$

$$= \arg\min_{\boldsymbol{w}} \ -\mathbf{p}^{\text{Adam}}(\boldsymbol{\theta})^T\boldsymbol{w} + \frac{\eta}{2}\boldsymbol{w}^T\mathbf{Q}^{\text{Adam}}(\boldsymbol{\theta})^T\boldsymbol{w} \tag{27}$$

Where $\mathbf{p}^{\text{Adam}}$ and $\mathbf{Q}^{\text{Adam}}$ are defined in Equations 8 and 9, respectively.

## D  Proof of Theorem 4.1

We begin by noting a property of the landmark-based approximation introduced in Section 4.3: it exhibits *rotational equivariance*. That is, if all source and target gradients are rotated by an orthonormal matrix, the resulting landmark-based gradient approximations will also be simply rotated by the same matrix.

In the remainder of this section, we prove two useful lemmas—Lemma D.1 and Lemma D.2. We then state and prove Theorem D.3, which bounds the error in the vector $\mathbf{p}$ for any unbiased approximation that satisfies rotational equivariance. Finally, Corollary D.4 bounds the difference in the resulting sample weights, thereby completing the proof of Theorem 4.1.

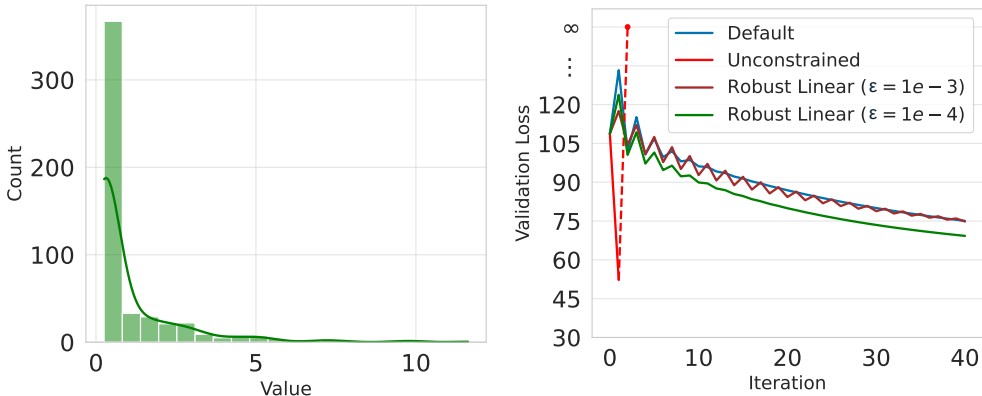

Figure 6: (Left) Distribution of theoretical robust weights for the linear case with $\epsilon = 10^{-4}$, and (Right) validation loss during training with different variants in the running experiment setting.

**Lemma D.1.** *Given unit vectors* $\mathbf{g}, \mathbf{t} \in \mathbb{R}^d$, *assume* $\hat{\mathbf{g}} = \mathbf{g} + \mathbf{e}$ *is a noisy approximation to* $\mathbf{g}$. *Additionally, assume* $\mathbf{e} \in \mathbb{R}^d$ *is a zero-mean, isotropic random vector, i.e.,* $\mathbb{E}[\mathbf{e}] = 0$ *and* $Cov(\mathbf{e}) = \sigma^2 \mathbf{I}_d$ *for some* $\sigma > 0$. *Let* $S = \langle \hat{\mathbf{g}}, \mathbf{t} \rangle$. *Then* $\mathbb{E}[S] = \langle \mathbf{g}, \mathbf{t} \rangle$, *and* $Var(S) = \frac{\mathbb{E}[||\hat{\mathbf{g}} - \mathbf{g}||_2^2]}{d}$.

*Proof.* The expectation of $S$ follows directly from zero-mean property of $\mathbf{e}$. To bound its variance, let $\Sigma$ denote the covariance matrix of $\mathbf{e}$. Since $\mathbf{e}$ is isotropic, $\Sigma = \sigma^2 \mathbf{I}_d$ for some $\sigma$. We can write:

$$
\begin{aligned}
\text{Var}(S) &= \text{Var}\langle \hat{\mathbf{g}}, \mathbf{t} \rangle \\
&= \text{Var}(\langle \mathbf{g}, \mathbf{t} \rangle + \langle \mathbf{e}, \mathbf{t} \rangle) \\
&= \text{Var}(\langle \mathbf{e}, \mathbf{t} \rangle) \\
&= \mathbf{t}^T \Sigma \mathbf{t} \\
&= \sigma^2 ||\mathbf{t}||^2 \\
&= \sigma^2
\end{aligned}
\tag{28}
$$

Also,

$$
\begin{aligned}
\mathbb{E}[||\hat{g} - g||^2] &= \mathbb{E}[||\mathbf{e}||^2] \\
&= tr(\Sigma) \\
&= d\sigma^2
\end{aligned}
\tag{29}
$$

Hence $\sigma^2 = \frac{\mathbb{E}[||\hat{g} - g||^2]}{d}$, which concludes the proof. $\qquad\square$

**Lemma D.2.** *Assume* $\mathbf{x} \in \mathbb{R}^d$ *is a random vector from an arbitrary distribution. For any random orthonormal matrix of the form* $\mathbf{R} = \mathbf{PD}$, *where*

- $\mathbf{P}$ *is a random permutation matrix*

- $\mathbf{D}$ *is a diagonal matrix with i.i.d. Rademacher signs* ($\pm 1$)

*the random vector* $\mathbf{y} = \mathbf{Rx}$ *is isotropic, i.e.,* $Cov(\mathbf{y}) = \sigma^2 \mathbf{I}_d$ *for some real value* $\sigma$.

*Proof.* We can write:

$$
\begin{aligned}
\text{Cov}(\mathbf{Rx}) &= \mathbb{E}_{\mathbf{P},\mathbf{D},\mathbf{x}}[\mathbf{Rxx}^T\mathbf{R}^T] \\
&= \mathbb{E}_{\mathbf{P},\mathbf{D},\mathbf{x}}[\mathbf{PDxx}^T\mathbf{DP}^T] \\
&= \mathbb{E}_{\mathbf{x}}[\mathbb{E}_{\mathbf{P}}[\mathbb{E}_{\mathbf{D}}[\mathbf{PDxx}^T\mathbf{DP}^T \mid \mathbf{P}, \mathbf{x}] \mid \mathbf{x}]] \\
&= \mathbb{E}_{\mathbf{x}}[\mathbb{E}_{\mathbf{P}}[\mathbf{P}\mathbb{E}_{\mathbf{D}}[\mathbf{Dxx}^T\mathbf{D} \mid \mathbf{x}]\mathbf{P}^T \mid \mathbf{x}]]
\end{aligned}
\tag{30}
$$

Now note that $\mathbb{E}_{\mathbf{D}}[\mathbf{D}\mathbf{x}\mathbf{x}^T\mathbf{D} \mid \mathbf{x}] = \text{diag}(\mathbf{x}^2)$. Substituting into the expectation over $\mathbf{P}$, we need to compute $\mathbb{E}_{\mathbf{P}}[\mathbf{P}\text{diag}(\mathbf{x}^2)\mathbf{P}^T \mid \mathbf{x}]$. However, since $\mathbf{P}$ is a random permutation, off-diagonal elements are zero and for the diagonal elements, any element of $\mathbf{x}^2$ can be picked with equal probability. Hence, the expectation over $\mathbf{P}$ equals $\frac{1}{d}||\mathbf{x}||^2\mathbf{I}_d$.

Putting it all together in Equation 30, we get

$$\text{Cov}(\mathbf{R}\mathbf{x}) = \frac{\mathbb{E}[||\mathbf{x}||^2]}{d}\mathbf{I}_d, \tag{31}$$

which concludes the proof. $\qquad\square$

**Theorem D.3.** *Let $\mathbf{G} \in \mathbb{R}^{n \times d}$ and $\mathbf{t} \in \mathbb{R}^d$, with $\mathbf{t}$ and each row of $\mathbf{G}$ having unit lengths. Let $\mathbf{g}_i$ denote the $i$'th row in $\mathbf{G}$. Additionally, assume access to a (randomized) mapping function $\mathbf{h} : \{\mathbf{g}_1, \mathbf{g}_2, \ldots, \mathbf{g}_n\} \to \mathbb{R}^d$, and let $\forall i \in \{1, 2, \ldots, n\} : \hat{\mathbf{g}}_i = \mathbf{h}(\mathbf{g}_i; \mathbf{G})$. Additionally, assume $\mathbf{h}(.)$ satisfies:*

1. *Unbiased: $\forall i \in \{1, 2, \ldots, n\} : \mathbb{E}[\hat{\mathbf{g}}_i] = \mathbf{g}_i$, i.e., $\mathbf{h}(.)$ is unbiased.*

2. *Bounded Average Mean Squared Error: Let $\delta_i^2 = \mathbb{E}[||\hat{\mathbf{g}}_i - \mathbf{g}_i||^2]$. Then:*

$$\frac{1}{n}\sum_{i=1}^{n}\delta_i^2 \le \Delta^2$$

   *for some $\Delta^2 \ge 0$.*

3. *Rotation Equivariance: For any orthonormal rotation matrix $\mathbf{R} \in \mathbb{R}^{d \times d}$ and $\forall i \in \{1, 2, \ldots, n\} : \mathbf{h}(\mathbf{R}\mathbf{g}_i; \mathbf{G}\mathbf{R}) = \mathbf{R}\hat{\mathbf{g}}_i$.*

*Construct the vector $\mathbf{p} = [p_1, p_2, \ldots, p_n]^T$ such that $p_i = \langle \mathbf{g}_i, \mathbf{t}\rangle$. Similarly define $\hat{\mathbf{p}} = [\hat{p}_1, \hat{p}_2, \ldots, \hat{p}_n]^T$, where $\hat{p}_i = \langle \hat{\mathbf{g}}_i, \mathbf{t}\rangle$. Then*

$$\mathbb{E}[||\mathbf{p} - \hat{\mathbf{p}}||^2] \le \frac{n\Delta^2}{d} \tag{32}$$

*Proof.* For all $i$, let $\mathbf{e}_i = \hat{\mathbf{g}}_i - \mathbf{g}_i$ denote the error. By the *Unibased* assumption, $\mathbb{E}[\mathbf{e}_i] = \mathbf{0}$.

Without loss of generality, we can assume that for all $i$, the vector $\mathbf{e}_i$ is isotropic, i.e., $\text{Cov}(\mathbf{e}_i)$ is a scalar multiple of the identity matrix. If this is not the case, we take advantage of Lemma D.2 and apply a change of variables: $\mathbf{G} \leftarrow \mathbf{G}\mathbf{R}$ and $\mathbf{t} \leftarrow \mathbf{R}\mathbf{t}$, where $\mathbf{R} = \mathbf{P}\mathbf{D}$, $\mathbf{P}$ is a permutation matrix, and $\mathbf{D}$ is a diagonal matrix with entries chosen uniformly at random from $\{\pm 1\}$. Note that by the *Rotation Equivariance* assumption, this transformation implies $\hat{\mathbf{g}}_i \leftarrow \mathbf{R}\hat{\mathbf{g}}_i$. Under this transformation, the error vectors $\mathbf{e}_i$ are mapped into a space where they become isotropic, and the pairwise dot products and distances remain unchanged as $\mathbf{R}$ is orthonormal.

Now we can directly apply Lemma $D.1$ for each coordinate $i$: $\mathbb{E}[\hat{p}_i] = p_i$ and $\text{Var}(\hat{p}_i) = \frac{\mathbb{E}[||\hat{g}-g||^2]}{d}$. This means:

$$\begin{aligned}
\mathbb{E}[||\mathbf{p} - \hat{\mathbf{p}}||^2] &= \sum_{i=1}^{n}\mathbb{E}[(p_i - \hat{p}_i)^2] \\
&= \sum_{i=1}^{n}\text{Var}(\hat{p}_i) \\
&= \sum_{i=1}^{n}\frac{\mathbb{E}[||\hat{g}-g||^2]}{d} \\
&\le \frac{n\Delta^2}{d}
\end{aligned}$$

where the last inequality comes from the *Bounded Average Mean Squared Error* assumption. $\qquad\square$

**Corollary D.4.** *In the setting of Theorem D.3, if we define:*

$$\boldsymbol{w}(\mathbf{p}) = \arg\min_{\mathbf{x}} -\mathbf{p}^T\mathbf{x} + \frac{\lambda}{2}||\mathbf{x}||_2^2, \ \ s.t. \ \begin{cases} \mathbf{x} \ge 0 \\ \mathbf{x}^T\mathbb{1} = n \end{cases} \tag{33}$$

*then*

$$\mathbb{E}[||\boldsymbol{w}(\mathbf{p}) - \boldsymbol{w}(\hat{\mathbf{p}})||^2] \leq \frac{n\Delta^2}{\lambda^2 d}. \tag{34}$$

*Proof.* Let $F_p(x) = -\mathbf{p}^T\mathbf{x} + \frac{\lambda}{2}||\mathbf{x}||_2^2$ and $C = \{\mathbf{x} \in \mathbb{R}^d : \mathbf{x} \geq 0, \mathbf{x}^T\mathbb{1} = n\}$. Note that the objective above has a unique solution since $F_p$ is $\lambda$-strongly convex and $C$ is a convex set independent of $\mathbf{p}$.

By strong convexity, $\forall x, y \in \mathbb{R}^d$:

$$F_p(\mathbf{y}) \geq F_p(\mathbf{x}) + \nabla_x F_p(\mathbf{x})^T(\mathbf{y} - \mathbf{x}) + \frac{\lambda}{2}||\mathbf{y} - \mathbf{x}||^2 \tag{35}$$

Set $\mathbf{x} = \boldsymbol{w} := \boldsymbol{w}(\mathbf{p})$ and $\mathbf{y} = \hat{\boldsymbol{w}} := \boldsymbol{w}(\hat{\mathbf{p}})$. Since $\boldsymbol{w}$ minimizes $F_p$ over $C$, $\nabla_x F_p(\mathbf{x})^T(\mathbf{y} - \boldsymbol{w}) \geq 0$. Hence:

$$F_p(\hat{\boldsymbol{w}}) \geq F_p(\boldsymbol{w}) + \frac{\lambda}{2}||\boldsymbol{w} - \hat{\boldsymbol{w}}||^2 \tag{36}$$

Swapping $\boldsymbol{w}$ and $\hat{\boldsymbol{w}}$,

$$F_{\hat{p}}(\boldsymbol{w}) \geq F_{\hat{p}}(\hat{\boldsymbol{w}}) + \frac{\lambda}{2}||\boldsymbol{w} - \hat{\boldsymbol{w}}||^2 \tag{37}$$

Adding the two equations above:

$$(\mathbf{p} - \hat{\mathbf{p}})^T(\boldsymbol{w} - \hat{\boldsymbol{w}}) \geq \lambda||\boldsymbol{w} - \hat{\boldsymbol{w}}||^2 \tag{38}$$

Applying Cauchy-Schwarz on the left hand side, we get

$$||\mathbf{p} - \hat{\mathbf{p}}||\cdot||\boldsymbol{w} - \hat{\boldsymbol{w}}|| \geq \lambda||\boldsymbol{w} - \hat{\boldsymbol{w}}||^2 \tag{39}$$

Hence

$$||\boldsymbol{w} - \hat{\boldsymbol{w}}|| \leq \frac{1}{\lambda}||\mathbf{p} - \hat{\mathbf{p}}|| \tag{40}$$

Combining with the result of Theorem D.3:

$$\mathbb{E}[||\boldsymbol{w} - \hat{\boldsymbol{w}}||^2] \leq \frac{n\Delta^2}{\lambda^2 d}. \tag{41}$$

$\square$

# E  Dataset and Model Details

This section provides details on the datasets and models used throughout the paper.

## E.1  Datasets

For the datasets, we largely follow the setup of Ivison et al. [2025].

**Tulu V2 (ODC-BY License).** The Tulu V2 dataset [Ivison et al., 2023], also known as the Tulu V2 SFT Mixture, is a comprehensive instruction-tuning dataset. Following Ivison et al. [2025], we consider the unfiltered version with 5.8M samples, consisting of 961,322 samples from FLAN v2 [Chung et al., 2024], 398,439 samples from FLAN CoT [Chung et al., 2024], 7,707 samples from Open Assistant [Köpf et al., 2023], 15,007 from Dolly [Conover et al., 2023], 52,002 from GPT-4 Alpaca [Peng et al., 2023], 20,022 from Code Alpaca [Chaudhary, 2023], 100,054 from ShareGPT, 1,030 from LIMA [Zhou et al., 2023b], 142,802 from Wizard Evol-Instruct V2 [Xu et al., 2023], 4,111,858 from Open Orca [Lian et al., 2023], 7,535 from SciRIFF [Wadden et al., 2024], and 14 from Hardcoded. For more information, we refer the reader to Ivison et al. [2025].

**MMLU (MIT License).** The Massive Multitask Language Understanding (MMLU) dataset [Hendrycks et al., 2021a,b] consists of challenging multiple-choice questions from 57 topics, such as abstract algebra, astronomy, machine learning, and more. It includes 5 development samples per category and a total of 14,042 test samples. We use the development samples as our target set and evaluate the final model zero-shot on the test set.

**GSM8K (MIT License).** This dataset comprises grade school math questions, with 7.47k training and 1.32k test samples [Cobbe et al., 2021]. We evaluate the models on the test set using 8 examples

in the context (8-shot evaluation) and use the same 8 individual samples as the target set. As is standard, only the final answer to each question is considered.

**Big-Bench-Hard (MIT License).** This dataset includes questions from 27 challenging tasks, such as causal judgment, multi-step arithmetic, and logic. Following Suzgun et al. [2022], we perform 3-shot evaluations using the same 3 samples per category (a total of 81) as the target set.

**TyDIQA (Apache-2.0 License).** TyDIQA is a dataset of 204k question-answering samples across 11 languages [Clark et al., 2020]. For evaluation, we follow Ivison et al. [2025], which in turn follows Anil et al. [2023], using 1-shot prompting. We select 9 samples per language for the target set.

**Codex (MIT License).** This dataset contains 164 Python programming questions [Chen et al., 2021], of which 16 are used as the target set and the remaining as the test set. See Ivison et al. [2025] for additional evaluation details.

**SQuAD (CC BY-SA 4.0 License).** The Stanford Question Answering Dataset (SQuAD) [Rajpurkar et al., 2016] contains reading comprehension questions based on Wikipedia articles. We use 500 random samples from the training split as the target set. We perform 3-shot evaluations with three samples randomly selected from the training set.

## E.2 Model Licenses

In this paper, we utilize LLaMA 2 [Touvron et al., 2023], LLaMA 3.2 3B [Grattafiori et al., 2024], Qwen 2.5 1.5B [Team, 2024], and Qwen 2.5 3B [Team, 2024] models. These models are distributed under the LLaMA 2 Community License, LLaMA 3.2 Community License, Apache-2.0 License, and Qwen Research License, respectively.

## F Embeddings Study

In Section 4.3, we noted that existing embedding functions are insufficient for our landmark-based gradient approximations and introduced the JVP embeddings as an alternative. In this section, we compare different embedding functions in two settings. In all the experiments, the model we consider is Llama-2 7B [Touvron et al., 2023].

**Gradient Recovery.** First, we randomly take 200k samples from Tulu V2 [Ivison et al., 2023] and embed them using various embedding functions. We then use a small number of landmark gradient samples (selected uniformly at random) to approximate the gradients for all data points, following the method described in Section 4.3. This process is repeated for different numbers of landmarks to evaluate how performance varies with landmark count. We report the average cosine similarity between the approximated gradients and the true gradients (projected into 8192-dimensional space using Rademacher-based projections [Ivison et al., 2025, Park et al., 2023]) for each case.

We evaluate several embedding functions: the RDS+ embeddings from Ivison et al. [2025], NVIDIA's NV-Embed-v2 [Lee et al., 2024], GTR-base [Ni et al., 2021], and our proposed JVP-based approach using two random vectors and four transformer blocks.

As a lower bound, we also include a *Trivial* embedding: here, we assume that the gradients for the landmark samples are perfectly recovered, while the gradients for all other samples are treated as completely random.

Figure 7 (Left) presents a comparison of these embedding functions. Our JVP embeddings outperform all other methods, including the more computationally intensive RDS+ and NV-Embed-v2.

Finally, we compute an upper bound by using the true projected gradients as the embedding function and repeating the same experiment. As shown in Figure 7 (Right), this idealized setting quickly achieves high accuracy in gradient approximation—surpassing 0.9 cosine similarity with just over 4096 landmarks. This suggests that the gradients are approximately low-rank, a known phenomenon in LLMs [Hu et al., 2022, Zhao et al., 2024].

**End-to-end Selection and Training.** We repeat the selection and fine-tuning experiments from Table 1, this time replacing the JVP embeddings with either GTR-base or true gradient embeddings. Table 3 reports the resulting accuracy for each task. Due to the high computational cost of obtaining true gradients, we include only a single random seed for this setting.

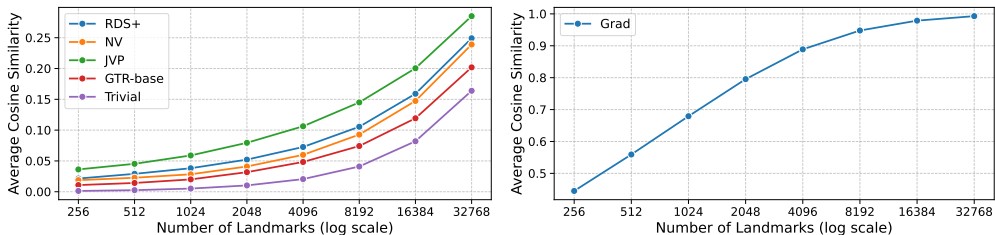

Figure 7: (Left) Gradient direction recovery vs number of landmarks, when different proxy embdeding functions are used, and (Right) gradient direction recovery when the actual gradients are used as an ideal embedding.

Table 3: Accuracy ($\pm$ standard deviation) of Llama2-7B across six tasks when using Influence Distillation with different embeddings to select 10k samples from a pool of 200k in the Tulu V2 dataset [Ivison et al., 2023]. The number of landmarks is fixed at 4096.

| Model | Embedding | MMLU | GSM8k | BBH | TyDIQA | CODEX | SQuAD | Avg. $\Delta$ w/ Uniform |
|---|---|---|---|---|---|---|---|---|
| | GTR-base | $46.7 \pm 0.17$ | $18.7 \pm 0.27$ | $42.8 \pm 0.34$ | $52.2 \pm 0.56$ | $29.3 \pm 0.84$ | $82.1 \pm 0.30$ | 45.3 |
| Llama2-7B | JVP | $48.3 \pm 0.21$ | $20.3 \pm 1.65$ | $43.2 \pm 0.67$ | $53.6 \pm 0.34$ | $29.5 \pm 3.14$ | $83.2 \pm 1.02$ | 46.4 |
| | Grad | 48.3 | 20.2 | 43.7 | 51.7 | 27.7 | 84.5 | 46.0 |

We fix the number of landmarks to 4096 across all experiments. The results show that while GTR-base consistently underperforms, the JVP and true gradient embeddings yield comparable accuracy—falling within each other's standard deviation in most cases. This indicates that the gradient approximations provided by JVP embeddings are sufficiently accurate for end-to-end training.

Finally, we note that since Figure 7 (Right) demonstrates near-perfect gradient recovery using the Grad embedding, the corresponding row in Table 3 closely mirrors the performance of the LESS method [Xia et al., 2024].

# G An Active-Set Solution

In this appendix we derive the solution to the Influence Distillation objective under the assumption that $\eta \mathbf{Q} + \lambda \mathbf{I}$ is positive definite (PD). This setting includes the special first-order case used in the main body of the paper, where $\eta \to 0$. Concretely, we solve

$$\boldsymbol{w}^* = \arg\min_{\boldsymbol{w}} \; -\mathbf{p}^T \boldsymbol{w} + \frac{\eta}{2} \boldsymbol{w}^T \mathbf{Q} \boldsymbol{w} + \frac{\lambda}{2} \boldsymbol{w}^T \boldsymbol{w}, \;\; s.t. \begin{cases} \boldsymbol{w} \geq 0 \\ \boldsymbol{w}^T \mathbb{1} = n \end{cases} \tag{42}$$

where $n$ denotes the dimension of $\boldsymbol{w}$ and $\eta \mathbf{Q} + \lambda \mathbf{I} \succ \mathbf{0}$.

Introduce the Lagrange multipliers $\tau \in \mathbb{R}$ for the equality constraint and $\boldsymbol{\alpha} \in \mathbb{R}^n_{\geq 0}$ for the non-negativity constraints. The Lagrangian is

$$L(\boldsymbol{w}, \tau, \boldsymbol{\alpha}) = -\mathbf{p}^\top \boldsymbol{w} + \frac{\eta}{2} \boldsymbol{w}^\top \mathbf{Q} \boldsymbol{w} + \frac{\lambda}{2} \boldsymbol{w}^\top \boldsymbol{w} - \tau(\mathbb{1}^\top \boldsymbol{w} - n) - \boldsymbol{\alpha}^\top \boldsymbol{w}. \tag{43}$$

Differentiating $L$ with respect to $\boldsymbol{w}$ and setting it equal to zero yields

$$\eta \mathbf{Q} \boldsymbol{w} + \lambda \boldsymbol{w} - \mathbf{p} - \tau \mathbb{1} - \boldsymbol{\alpha} = \mathbf{0}. \tag{44}$$

Let $\mathbf{R} := \eta \mathbf{Q} + \lambda \mathbf{I} \succ \mathbf{0}$. Then

$$\mathbf{R} \boldsymbol{w} - \mathbf{p} - \tau \mathbb{1} - \boldsymbol{\alpha} = \mathbf{0}. \tag{45}$$

By complementary slackness, $\forall i : \boldsymbol{w}_i \boldsymbol{\alpha}_i = 0$. Let $A = \{i \mid w_i = 0\}$ be the active set and $B$ its complement. Restricting (45) to the free indices gives

$$\mathbf{R}_{BB} \boldsymbol{w}_B = \mathbf{p}_B + \tau \mathbb{1}_B. \tag{46}$$

Because $\mathbf{R}_{BB}$ is a principal sub-matrix of the PD matrix $\mathbf{R}$, it is itself PD. Hence

$$\boldsymbol{w}(\tau; B) = \mathbf{R}_{BB}^{-1}(\mathbf{p}_B + \tau \mathbb{1}_B). \tag{47}$$

Enforcing $\mathbb{1}^T \boldsymbol{w} = n$ determines $\tau$:

$$\mathbb{1}_B^T \mathbf{R}_{BB}^{-1}(\mathbf{p}_B + \tau \mathbb{1}_B) = n, \tag{48}$$

and therefore

$$\tau^* = \frac{n - \mathbb{1}_B^T \mathbf{R}_{BB}^{-1} \mathbf{p}_B}{\mathbb{1}_B^T \mathbf{R}_{BB}^{-1} \mathbb{1}_B}. \tag{49}$$

Substituting $\tau^*$ back into $\boldsymbol{w}(\tau; B)$ gives us the weights on $B$:

$$\boldsymbol{w}_B^* = \mathbf{R}_{BB}^{-1}\Big(\mathbf{p}_B + (\frac{n - \mathbb{1}_B^T \mathbf{R}_{BB}^{-1} \mathbf{p}_B}{\mathbb{1}_B^T \mathbf{R}_{BB}^{-1} \mathbb{1}_B})\mathbb{1}_B\Big). \tag{50}$$

For indices in the active set $A$ we have $\boldsymbol{w}_A^* = \mathbf{0}$, giving the final candidate solution $\boldsymbol{w}^* = (\boldsymbol{w}_A^*, \boldsymbol{w}_B^*)$.

Optimality requires that the remaining Karush–Kuhn–Tucker (KKT) conditions hold, namely $\forall i \in B$, $\boldsymbol{w}_i \geq 0$ (primal feasibility) and $\forall j \in A$, $\boldsymbol{\alpha}_j \geq 0$ (dual feasibility). Because the objective is convex ($\mathbf{R} \succ 0$), any partition $A, B$ satisfying these conditions is the global optimum.

Examining the coordinates in $A$ in (45) gives

$$\boldsymbol{\alpha}_A = (\mathbf{R}_{AB}\boldsymbol{w}_B^*)_A - \mathbf{p}_A - \tau^* \mathbb{1}_A. \tag{51}$$

Problems of this type are typically solved with a primal–dual active-set algorithm. We start from the feasible point $\boldsymbol{w} = \mathbb{1}$ (so $A = \varnothing$, $B = \{1, \ldots, n\}$) and repeat:

1. Solve for $\boldsymbol{w}_B^*$ via (50).

2. If any component of $\boldsymbol{w}_B^*$ is negative, move its index to $A$.

3. Compute $\boldsymbol{\alpha}_A$; if any component is negative, move its index back to $B$.

Each move strictly decreases the objective, and with only finitely many index sets the algorithm terminates once all components of $\boldsymbol{w}_B$ and $\boldsymbol{\alpha}_A$ are non-negative.

**The Special Case of $\eta \to 0$.** This setting corresponds to the first-order Influence Distillation variant used throughout the main body of the paper. In this case, we demonstrate that as $\lambda$ increases, the solution $\boldsymbol{w}^*$ becomes denser—that is, it contains more non-zero elements. This observation is leveraged in Section 4.4 for tuning the parameter $\lambda$.

When $\eta \to 0$, we can write $\mathbf{R} = \lambda \mathbf{I}$, which implies $\mathbf{R}_{BB}^{-1} = \frac{1}{\lambda}\mathbf{I}$ and $\mathbf{R}_{AB} = \mathbf{0}$. Substituting these into Equations 49, 50, and 51, we obtain:

$$\tau^* = \frac{n\lambda - \mathbb{1}_B^T \mathbf{p}_B}{|B|} \tag{52}$$

$$\boldsymbol{w}_B^* = \frac{1}{\lambda}(\mathbf{p} + \tau^* \mathbb{1})_B \tag{53}$$

$$\boldsymbol{\alpha}_A = -(\mathbf{p} + \tau^* \mathbb{1})_A \tag{54}$$

Since both $\boldsymbol{w}_B$ and $\boldsymbol{\alpha}_A$ must be non-negative, the last two equations imply that the active set $B$ must satisfy $B = \{i : \mathbf{p}_i \geq -\tau^*\}$, i.e., $B$ is necessarily a set of top-k elements from $\mathbf{p}$ for some $k$.

Consider two values $\lambda_1 < \lambda_2$, and let $B_1$ and $B_2$ denote their optimal supports with sizes $k_1$ and $k_2$, and $\boldsymbol{w}^{(1)}$ and $\boldsymbol{w}^{(2)}$ their respective optimal weight vectors; similarly, let $\boldsymbol{\alpha}^{(1)}$ and $\boldsymbol{\alpha}^{(2)}$ denote their associated dual variables. Suppose for contradiction that $k_2 < k_1$. Note that $B_1$ consists of the indices of the top $k_1$ elements in $\mathbf{p}$, while $B_2 \subset B_1$ includes the top $k_2$ elements of $\mathbf{p}$. Let $s_{k_1}$ and $s_{k_2}$ represent the sums of the top $k_1$ and $k_2$ elements in $\mathbf{p}$, respectively. Define $j$ as the index of the $k_1$-th largest element in $\mathbf{p}$. Since $j \in B_1$, we have $\boldsymbol{w}_j^{(1)} \geq 0$, and since $j \notin B_2$, it follows that

$\boldsymbol{\alpha}_j^{(2)} \geq 0$. Therefore,

$$\boldsymbol{w}_j^{(1)} \geq 0$$

$$\Rightarrow \mathbf{p}_j + \frac{n\lambda_1 - s_{k_1}}{k_1} \geq 0$$

$$\Rightarrow n\lambda_1 \geq s_{k_1} - k_1 \mathbf{p}_j = \sum_{i \in B_1} (\mathbf{p}_i - \mathbf{p}_j)$$

and

$$\boldsymbol{\alpha}_j^{(2)} \geq 0$$

$$\Rightarrow \mathbf{p}_j + \frac{n\lambda_2 - s_{k_2}}{k_2} \leq 0$$

$$\Rightarrow n\lambda_2 \leq s_{k_2} - k_2 \mathbf{p}_j = \sum_{i \in B_2} (\mathbf{p}_i - \mathbf{p}_j)$$

Observe that $\sum_{i \in B_2} (\mathbf{p}_i - \mathbf{p}_j) \leq \sum_{i \in B_1} (\mathbf{p}_i - \mathbf{p}_j)$ by definition of $\mathbf{p}_j$, leading to the inequality $n\lambda_2 \leq n\lambda_1$, which contradicts our initial assumption that $\lambda_1 < \lambda_2$.

This contradiction confirms that as the regularization parameter $\lambda$ increases, the solution becomes progressively denser. Specifically, at $\lambda = 0$, the solution concentrates all weight on the largest element of $\mathbf{p}$ to minimize the objective, whereas in the limit as $\lambda \to \infty$, the regularization dominates, resulting in $\boldsymbol{w} = \mathbb{1}$.

## H  First- vs Second-Order Influence Distillation

Recall the robust Objective 7

$$\boldsymbol{w}^* = \underset{\boldsymbol{w}}{\arg\min} \ f(\boldsymbol{w}; \boldsymbol{\theta}) + \frac{\lambda}{2} \|\boldsymbol{w}\|_2^2, \ \ s.t. \ \begin{cases} \boldsymbol{w} \geq 0 \\ \boldsymbol{w}^T \mathbb{1} = |S| \end{cases} \tag{55}$$

where,

$$f(\boldsymbol{w}; \boldsymbol{\theta}) = -\mathbf{p}(\boldsymbol{\theta})^T \boldsymbol{w} + \frac{\eta}{2} \boldsymbol{w}^T \mathbf{Q}(\boldsymbol{\theta}) \boldsymbol{w} \tag{56}$$

In this section, we compare the first-order term $T_1 = \mathbf{p}(\boldsymbol{\theta})^T \boldsymbol{w}$ with the second-order term $T_2 = \frac{\eta}{2} \boldsymbol{w}^T \mathbf{Q}(\boldsymbol{\theta}) \boldsymbol{w}$. To do so, we sample 128 random examples from the Tulu V2 dataset [Ivison et al., 2023] as the source dataset, and 4 examples from either GSM8k [Cobbe et al., 2021] or MMLU [Hendrycks et al., 2021a,b] as the target dataset.

We compute the vectors $\mathbf{p}$ and the matrices $\mathbf{Q}$ exactly for the Qwen-2.5 1.5B model [Team, 2024], using Hessian-vector products to obtain $\mathbf{Q}$. We then evaluate both $T_1$ and $T_2$ using default weights $\boldsymbol{w} = \mathbb{1}$ and a range of learning rates. To measure the relative contribution of the second-order term, we report the ratio $\left| \frac{T_2}{T_1} \right|$.

As shown in Figure 8, the second-order term is generally negligible for practical learning rates ($\eta \leq 10^{-4}$), indicating that the first-order approximation is sufficient in this setting.

## I  Projection Details

While in some of our lower-cost experiments we employ Rademacher-based projections—including projecting JVP embeddings to a 4096-dimensional space using this method, as supported on GPUs by Park et al. [2023]—we find that projecting the landmark gradients with Rademacher projections becomes a computational bottleneck. To address this, we instead use a combination of pre-masking and Randomized Hadamard Transform-based projections, as described below.

**Hadamard-based Projection.** Given a high-dimensional gradient vector $\mathbf{g}$, we first pad it with zeros to the nearest power of two, $2^k$. Then, we apply a random sign ($\pm 1$) to each element. The signed vector is reshaped into a matrix $\mathbf{X}$ of dimensions $m = 2^{\lceil \frac{k}{2} \rceil}$ and $n = 2^{\lfloor \frac{k}{2} \rfloor}$. We then apply

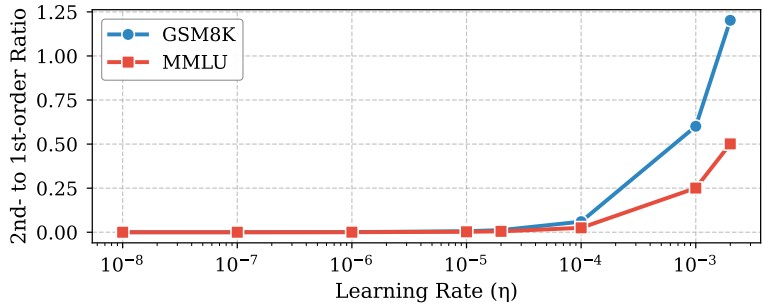

Figure 8: Ratio of second- to first-order terms for Qwen-2.5 1.5B across learning rates on two target datasets.

Table 4: Accuracy (± standard deviation) of Llama2-7B across six tasks when using Influence Distillation to select 10k samples from a pool of 200k in the Tulu V2 dataset [Ivison et al., 2023], with and without loss weighting during training. The number of landmarks is fixed at 8192.

| Model | Embedding | MMLU | GSM8k | BBH | TyDIQA | CODEX | SQuAD | Avg. Δ w/ Uniform |
|-------|-----------|------|-------|-----|--------|-------|-------|-------------------|
| Llama2-7B | Weighted | $47.8 \pm 0.16$ | $19.5 \pm 0.06$ | $42.3 \pm 0.26$ | $52.2 \pm 1.38$ | $27.0 \pm 2.53$ | $84.4 \pm 0.48$ | +1.48 |
| | Not Weighted | $48.2 \pm 0.35$ | $19.6 \pm 0.79$ | $42.4 \pm 0.14$ | $52.7 \pm 1.67$ | $29.3 \pm 1.27$ | $83.4 \pm 0.86$ | +1.93 |

Hadamard transforms from both sides: $\mathbf{H}_m^T \mathbf{X} \mathbf{H}_n$. The resulting matrix is flattened, and a random subset of its entries is selected as the projected vector.

Importantly, both the random sign patterns and the final index subset are generated once and reused across all projected vectors. This ensures consistency and enables meaningful comparison. The left and right Hadamard transforms are highly efficient and provide strong mixing across rows and columns.

**Pre-masking.** Although efficient GPU implementations of the Hadamard transform exist [Agarwal et al., 2024, Dao, 2023], they support transforms up to dimension $2^{15} = 32,768$. This allows us to efficiently project vectors of up to $2^{30} = 1,073,741,824$ elements—just over one billion. However, the full gradients of large language models (LLMs) can exceed this size.

To address this, we apply *pre-masking*: we randomly select one billion elements from the gradient vector before projection. For LLaMA-2 7B [Touvron et al., 2023], we select these elements from the `down_proj` matrices, which we find to represent the overall gradients well. For smaller models, we randomly sample one billion elements from the entire gradient vector.

# J   Weighted Training Loss

In this section, we investigate the effect of incorporating the weights derived by Influence Distillation into the training loss. Specifically, we conduct an experiment using LLaMA-2 7B [Touvron et al., 2023], with a pool size of 200k and 8192 landmarks sampled from Tulu V2 [Ivison et al., 2023]. During training, we scale the loss of each selected sample by its corresponding weight.

Table 4 compares this weighted training setup with a baseline where the weights of the selected samples are ignored. The results show that incorporating weights during training does not improve performance—and in some cases, it may even degrade it. This may be due to some samples having near-zero weights, effectively pruning them from the training process.

# K   Additional Studies

## K.1   Analysis of Training Progression on MMLU

For a more granular analysis, we repeated the Llama2-7B experiment on MMLU to compare the accuracies of Influence Distillation, Uniform sampling, and RDS+ throughout training. In this

experiment (using a single seed), model checkpoints were saved every 25 steps to track performance over time. The MMLU accuracies at these intervals are reported in Table 5.

Table 5: MMLU Accuracy Progression for Llama2-7B by Method.

| Method | Step 25 | Step 50 | Step 75 | Step 100 | Step 125 | Step 150 |
|--------|---------|---------|---------|----------|----------|----------|
| Uniform | 44.7 | 45.4 | 45.8 | 46.2 | 46.2 | 46.1 |
| RDS+ | 45.1 | 45.3 | 46.0 | 45.9 | 46.2 | 46.2 |
| InfDist | 45.2 | 46.1 | 47.9 | 48.3 | 48.4 | 48.5 |

We also conducted a similar experiment using full training on the entire dataset. Our findings indicate that full training requires around 1100 steps to reach an accuracy of 48.5. In contrast, Influence Distillation achieves this same accuracy at step 150, representing a more than $7\times$ speedup in training efficiency.

### K.2 Investigation of Weight Transferability.

To investigate the transferability of weights across tasks, we compute sample weights for different target sets as described in the paper and subsequently analyze their pairwise Pearson correlations. This analysis utilizes the Llama2-7B [Touvron et al., 2023] model with 1024 landmarks and a data pool of 200k examples.

The the computed weights for four of the six tasks, presented in Table 6, exhibit high correlation $(0.5 - 0.8)$. This suggests that sample weights computed for one task may transfer effectively to others.

Table 6: Pairwise Pearson correlations of sample weights across tasks.

| | MMLU | GSM8k | BBH | TyDiQA | Codex | SQuAD |
|--------|---------|---------|---------|---------|---------|---------|
| MMLU | +1.0000 | -0.0430 | -0.5430 | +0.6953 | +0.7422 | +0.7734 |
| GSM8k | +0.9961 | +1.0000 | +0.2520 | +0.2188 | +0.2637 | -0.0938 |
| BBH | -0.5430 | +0.2520 | +1.0000 | -0.2021 | -0.2295 | -0.6719 |
| TyDiQA | +0.6953 | +0.2188 | -0.2021 | +1.0000 | +0.5938 | +0.5312 |
| Codex | +0.7422 | +0.2637 | -0.2295 | +0.5938 | +1.0000 | +0.5547 |
| SQuAD | +0.7734 | -0.0938 | -0.6719 | +0.5312 | +0.5547 | +1.0000 |

To further assess cross-task generalization, we use the sample weights computed for MMLU to select a 10k subset from the training pool. We then train a model on this subset and evaluate it on each of the remaining tasks. The results, comparing Influence Distillation and uniform sampling (averaged over three seeds), are presented in Table 7.

Table 7: Cross-task generalization performance. Samples are selected using MMLU weights.

| Method | MMLU | GSM8k | BBH | TyDiQA | Codex | SQuAD |
|---------|------|-------|------|--------|-------|-------|
| Uniform | 45.6 | 17.5 | 41.8 | 51.6 | 27.0 | 80.8 |
| InfDist | 48.2 | 16.8 | 42.6 | 50.6 | 28.2 | 82.1 |

These results show that, even though the samples were selected based on MMLU, Influence Distillation still outperforms uniform selection in most tasks when evaluated for three seeds.

### K.3 Ablation Study on Target Set Size

To study the effect of the target set size, we vary the number of samples per category in the MMLU target distribution. MMLU contains 57 categories. While the main paper uses 5 samples per category (285 samples in total) as the target set, this ablation repeats the Llama2-7B [Touvron et al., 2023] experiment using 1, 2, 3, and 4 samples per category. This results in target set sizes of 57, 114, 171, and 228, respectively.

The MMLU accuracies for each target set size are presented in Table 8.

Table 8: Effect of Target Set Size on MMLU Accuracy.

| Target Size | MMLU Accuracy |
|---|---|
| 57 | 48.1 |
| 114 | 48.6 |
| 171 | 48.2 |
| 228 | 48.1 |
| 285 | 48.5 |

These results suggest that Influence Distillation is relatively robust to the size of the target dataset in this setting, demonstrating stable performance across the tested range of target set sizes.

### K.4 Impact of Landmark Selection

We repeat the experiments from Appendix Figure 7 (Left) to compare different landmark selection methods. Specifically, we fix the number of landmarks to 4096 and compute the average cosine similarity of the landmark-based approximated gradients with true gradients when using the following landmark selection methods:

1. Uniformly at random
2. Leverage score-based sampling (in the JVP embeddings space)
3. Longest samples
4. Shortest samples

The results for the Llama2-7B [Touvron et al., 2023] model are presented in Table 9.

Table 9: Gradient recovery by landmark selection method.

|  | Uniform | Leverage | Shortest | Longest |
|---|---|---|---|---|
| Gradient recovery | 0.105 | 0.105 | 0.035 | 0.046 |

These results show that deliberately choosing the longest or shortest samples significantly degrades gradient approximation quality. On the other hand, choosing landmarks uniformly at random is on par with leverage score-based sampling, as mentioned in the Hyperparameters paragraph in Section 5.1.

### K.5 Transferability of JVP Embeddings Between Models

To investigate JVP embedding transferability, we conducted an experiment to study how the average cosine similarity between approximated and true gradients of Llama2-7B [Touvron et al., 2023] is affected by the choice of embedding model.

Specifically, we compare the performance of the following embeddings, fixing the number of landmarks at 4096:

- JVP from Llama2-7B (same as the target model)
- JVP from Llama3.2-3B [Grattafiori et al., 2024]
- JVP from Qwen2.5-1.5B [Team, 2024]
- JVP from Qwen2.5-3B [Team, 2024]

The results are summarized in Table 10 and compared against the recovery achieved by RDS+ [Ivison et al., 2025].

Comparing these results with the recovery achieved by RDS+, we conclude that JVP embeddings are highly transferable across models. Notably, JVP embeddings computed from Qwen2.5-3B approximate the gradients of Llama2-7B with similar quality to those obtained directly from Llama2-7B itself.

Table 10: Gradient recovery using JVP embeddings from different models.

| | JVP (Llama2-7B) | JVP (Llama3.2-3B) | JVP (Qwen2.5-1.5B) | JVP (Qwen2.5-3B) | RDS+ (Llama2-7B) |
|---|---|---|---|---|---|
| Gradient recovery | 0.105 | 0.098 | 0.093 | 0.103 | 0.073 |

Figure 9: Average gradient cosine similarity on unseen samples from GSM8k (top) and SQuAD (bottom) across checkpoints.

# L    Differed Figures

Figure 9 is included in this section, having been moved from Appendix A due to its size to improve readability.

