# OpenReview forum: "Efficient Data Selection at Scale via Influence Distillation"
_NeurIPS.cc/2025/Conference — NeurIPS 2025 poster_

### Official Review · Reviewer_a4sj · 2025-06-03

**Clarity:** 3
**Significance:** 3
**Originality:** 2
**Rating:** 4
**Confidence:** 4

**Summary:**

This paper introduces Influence Distillation, a novel mathematically-justified framework for efficient data selection in large language model (LLM) training. By leveraging second-order information (gradients and Hessians), the method optimally weights training samples based on their influence on a target distribution, with derivations provided for both Gradient Descent and Adam optimizers. To address scalability, a landmark-based approximation is proposed, where influence is computed for a small subset of landmark samples and propagated to all others using Jacobian-vector product embeddings, significantly reducing computational costs. Experiments on the Tulu V2 dataset with Llama and Qwen family models across tasks like MMLU, GSM8k, and SQuAD show that Influence Distillation matches or outperforms state-of-the-art methods while achieving up to 3.5× faster selection, positioning it as an efficient solution on the Pareto frontier of cost versus accuracy.

**Questions:**

1. The paper lacks comparisons with the following baselines: LESS[1], DSIR[2], QUAD[3], TAROT[4] and IFD[5]. Specifically, LESS replaces the influence function with gradient similarity to approximate data utility. QUAD employs the K-FAC method and data clustering to accelerate the computation of inverse Hessian-vector products (iHVP).  Superfiltering performs data selection based solely on loss metrics such as perplexity (PPL) or IFD. Please compare these methods in the experiments and explain the results in terms of both effectiveness and efficiency.

[1] LESS: Selecting influential data for targeted instruction tuning.

[2] Data selection for language models via importance resampling.

[3] Harnessing Diversity for Important Data Selection in Pretraining Large Language Models.

[4] TAROT: Targeted Data Selection via Optimal Transport.

[5] Superfiltering: Weak-to-strong data filtering for fast instruction-tuning.

2. The paper lacks an ablation study on critical hyperparameters, such as the optimal size of the "small subset of landmarks" during LLM instruction tuning—specifically, how many samples comprise this subset. This omission leaves users uncertain about efficiently determining this value when applying the method, as well as whether the optimal number remains consistent across different models and datasets.

3. To the best of my knowledge, there are several existing baselines that are closely related to your method, including:

i) IDEAL: Data Equilibrium Adaptation for Multi-Capability Language Model Alignment

ii) Harnessing Diversity for Important Data Selection in Pretraining Large Language Models

iii) MATES: Model-Aware Data Selection for Efficient Pretraining with Data Influence Models

iv) TAROT: Targeted Data Selection via Optimal Transport

Specifically, LESS improves effectiveness by computing gradient similarity at each training iteration. QUAD and IDEAL employs the K-FAC method to accelerate the computation of inverse Hessian-vector products (iHVP). TAROT leverages optimal transport in the gradient space to measure distances between data points. Please provide a detailed comparison(Only a theoretical explanation is needed) of your method against these approaches, highlighting both advantages and limitations.

**Ethical Concerns:**

["NO or VERY MINOR ethics concerns only"]

**Final Justification:**

The rebuttal has largely addressed my concerns, and I raise my score to 4.

**Limitations:**

yes.

**Paper Formatting Concerns:**

no.

**Quality:**

2

**Strengths And Weaknesses:**

Strengths:

1. The paper provides a mathematically justified framework for data selection using second-order information (gradients and Hessians), offering closed-form solutions for optimal sample weights under both Gradient Descent and Adam optimizers. This theoretical foundation distinguishes it from heuristic-based methods .

2. The landmark-based approximation significantly reduces computational overhead by leveraging a small subset of "landmark" samples and propagating their influence via Jacobian-vector product (JVP) embeddings. This makes the method feasible for large datasets.

3. Extensive experiments on multiple LLMs (Llama, Qwen) and diverse tasks (MMLU, GSM8k, SQuAD) demonstrate that Influence Distillation outperforms or matches state-of-the-art methods.

Weaknesses:

1. The experimental section includes relatively few baselines and lacks an ablation study on key parameters (see question 1 and 2).

2. There is a lack of discussion on other influence function-based methods for data selection in LLMs (see question 3).

---

> ### Author Rebuttal · Authors · 2025-07-31
>
> We thank the reviewer for sharing their valuable insights. Below find our response to the questions:
>
> > 1. The choice of number of landmarks
>
> We appreciate the reviewer’s comment. As shown in Figure 3 (left), we have empirically studied the effect of the number of landmarks on the performance of Llama2-7B. The results indicate that performance stabilizes around 4096 landmarks. For other models, we did not perform any tuning and simply used the same value (4096) across the board.
>
> Determining the optimal number of landmarks is indeed an important and interesting question. One potential approach is to incrementally add landmarks and, at each step, approximate the gradient of each landmark using the other ones. Once the approximation quality reaches a certain threshold, the process can be stopped and that number can be used as the chosen landmark count.
>
> We explored this idea empirically by implementing a procedure to adaptively select landmarks for Llama2-7B. The method proceeds as follows:
>
> 1. Begin with a small set of landmarks L, initialized with 32 randomly selected samples. Compute and project their gradients using a Rademacher-based projection (similar to LESS [1]).
>
>
> 2. For each sample in L, approximate its gradient as a linear combination of the other samples (**excluding itself**). We use kernel ridge regression (KRR) with an RBF kernel to perform this approximation. To ensure that each sample is not used in approximating itself, we set the diagonal of the Gram matrix to zero.
>
>
> 3. Compute and report the average cosine similarity between the true and approximated gradients.
>
>
> 4. If the size of L is less than 8192, add one new random sample to L, compute and project its gradient, and repeat from step 2. Otherwise, stop the procedure.
>
>
> Running this experiment, we observe that the average cosine similarity improves smoothly as the number of landmarks increases. The table below presents the average cosine similarity values for several different sizes of the landmark set L:
> | Size of L|32|64|128|256|512|1024|2048|4096|8192|
> |-|--|--|---|---|---|----|----|----|----|
> |Gradient recovery|0.1106|0.2317|0.3416|0.4534|0.5586|0.6797|0.7958|0.8894|0.9480|
>
> These results suggest that setting a threshold of 0.8-0.9 for approximation quality would naturally lead to selecting a number of landmarks consistent with what we used in the paper.
>
> Importantly, this procedure does not incur additional computational cost, as the gradients of the landmarks must be computed anyway for Influence Distillation.
>
> We thank the reviewer for this insightful and valuable suggestion. While we believe that this method needs further investigation to establish its effectiveness, we agree that including this or a similar approach for selecting the number of landmarks would strengthen the paper.
>
> [1] https://arxiv.org/pdf/2402.04333
>
> > 2. Discussion on other baselines
>
> We thank the reviewer for their constructive feedback and for suggesting several relevant baselines for comparison and discussion. This will help better position our work and clarify its contributions. We provide a point-by-point response below. We will include these results and discussions in the next revision of the paper.
>
> - IFD is a heuristic-based filter that relies on metrics like perplexity and is independent of the target model's training dynamics. Previous work (e.g., the RDS+ paper [6]) has already shown that such heuristics are generally outperformed by more sophisticated data selection methods like RDS+ with a large margin. Given our focus on model-aware influence and our competitiveness with state-of-the-art data selection methods, we believe a direct experimental comparison would be less informative for demonstrating the specific advantages of our approach.
>
> - DSIR is a model-agnostic method that uses importance resampling and does not leverage the target model’s parameters for selection. This places it in a different category from model-aware influence function methods like ours. This distinction is already discussed in the Related Work section.
>
> - MATES trains a small, separate model to predict the influence of data points during the main model's training. A key limitation is the substantial cost and complexity of training this auxiliary model, which must be done repeatedly. By contrast, our Influence Distillation technique is a far more streamlined framework that requires no auxiliary models.
>
>
> - IDEAL is an interesting concurrent work (published after the NeurIPS deadline), which _addresses a different problem_: Its goal is to find the optimal mixing of data from different domains or sources. Our method, in contrast, selects the most influential individual data points from a given data pool.
>
> - QUAD & TAROT & LESS: We agree that QUAD, TAROT, and LESS are relevant baselines. TAROT and LESS, however, require computing the gradients for each sample in the pool, which means they are significantly slower than Influence Distillation due to our efficient landmark-based gradient approximation. QUAD, on the other hand, though closely related, focuses on pre-training settings, which differs from our focus on supervised fine-tuning. Due to the significant computational resources and time required to implement and run these methods fairly, we only have included a comparison with LESS in the next point below. We will include comparisons with QUAD and TAROT as well in the next revision of our paper.
>
> Overall, this detailed examination of prior/concurrent work suggests that our original submission already compares against the most relevant baselines, and does not impact our claims of novelty or performance. We thank the reviewer again for their detailed additional references, which we will incorporate into the next revision of our work.
>
> [6] https://arxiv.org/pdf/2503.01807
>
> > 3. Comparison with LESS [1]
>
> We thank the reviewer for this valuable comment. As noted in the last paragraph of Appendix F (also mentioned in Section 5.1, Baselines), we have included a brief discussion of the LESS method. Specifically, as an ablation, we replaced JVP embeddings with actual projected gradients. Figure 7 (right) in the appendix shows that this yields a gradient recovery of 0.9 (in terms of average cosine similarity) with 4096 landmarks, making the approach, in this case, very similar to LESS which uses exact (projected) gradients. The last row of Table 2 shows that this variant performs on par with Influence Distillation using JVP embeddings.
>
> To provide a more complete comparison, we also ran the 10k/200k selection experiment on Llama2-7B using the LESS method (with one seed due to its high computational cost). In this setup, gradients are computed after a 10k-step warmup, projected down to 8192 dimensions, and compared to the target gradients. The top 10k examples are then selected based on similarity—following the original LESS procedure. Below we present a comparison between uniform sampling, InfDist with JVP embeddings, and LESS.
>
> |Method|MMLU|GSM8k|BBH|TyDIQA|CODEX|SQuAD|Avg|
> |------|----|-----|---|------|-----|-----|----|
> |Uniform|45.6±0.43|17.5±1.08|41.8±0.20|51.6±0.38|27.0±0.60|80.8±1.05|44.1|
> |InfDist+JVP Embedding (ours)|48.3±0.21|20.3±1.65|43.2±0.67|53.6±0.34|29.5±3.14|83.2±1.02|46.4|
> |LESS|48.2|22.7|43.4|55.8|29.1|86.0|47.5|
>
> While LESS achieves higher accuracy on average, it is significantly more computationally expensive. Our method requires approximately 872 TeraFLOPs (TFs) for embedding and selection, whereas LESS requires around 8400 TFs—approximately 10x more.
>
> We thank the reviewer again for this insightful suggestion and will include these results in the main body of the next revision.
>
> [1] https://arxiv.org/pdf/2402.04333

---

> > ### Author Response · Authors · 2025-08-04
> >
> > As the discussion period is coming to an end soon, we would like to kindly ask the reviewer to take a moment to review our rebuttal. We have included new results and additional discussion, and we hope we have addressed your concerns. We are happy to discuss any remaining questions or feedback.

---

> > > ### Comment · Reviewer_a4sj · 2025-08-05
> > >
> > > Sorry for the late reply. I recognize the efforts made by the authors, and the rebuttal has largely addressed my concerns.

---

### Official Review · Reviewer_MsKh · 2025-06-10

**Clarity:** 2
**Significance:** 2
**Originality:** 2
**Rating:** 4
**Confidence:** 3

**Summary:**

This paper introduces Influence Distillation, a new data selection framework for efficiently training Large Language Models (LLMs). The method uses second-order information to assign model-specific weights to training samples, which helps guide LLM fine-tuning towards strong performance on a target domain. To achieve scalability and reduce computational costs, Influence Distillation employs a landmark-based approximation where influence is calculated for a small subset of "landmark" samples and then efficiently extended to other samples. Experiments on the Tulu V2 dataset, covering tasks like GSM8k, SQUAD, and MMLU, demonstrate that Influence Distillation performs comparably to or better than state-of-the-art methods while accelerating selection by up to 3.5 times. The framework has been validated across various models from the Llama and Qwen families.

**Questions:**

N/A

**Ethical Concerns:**

["NO or VERY MINOR ethics concerns only"]

**Final Justification:**

The author provide sufficient explanation to my questions. I will uphold my score.

**Limitations:**

Yes

**Quality:**

2

**Strengths And Weaknesses:**

Strengths:

1. The method is computationally efficient and scalable, especially for large datasets, by using a landmark-based approximation. This allows influence to be precisely computed for a small subset of "landmark" samples and then efficiently propagated to all other samples.

2. Experiments show that Influence Distillation matches or outperforms state-of-the-art performance while achieving up to 3.5x faster selection. It also achieves higher performance compared to other more expensive selection baselines in three out of four models and remains competitive in the fourth.

3. It is a mathematically justified framework for data selection.


Weaknesses:

1. The unconstrained version of the proposed solution may produce negative or highly irregular sample weights, including excessively large values, which lack intuitive interpretation. And the unconstrained weights may overfit to the current set of model parameters and the target dataset.

2. The cost of the warm-up phase is excluded from runtime measurements, with the justification that it becomes negligible for larger training pools or can be shortened. However, this is an exclusion from the reported runtime.

3. Calculating the Hessian (Q(θ)) exactly requires storing the backward graph, which can incur extra memory costs. Constructing the G_S matrix requires computing gradients for each individual training sample, which is computationally expensive and memory-intensive

---

> ### Author Rebuttal · Authors · 2025-07-31
>
> We thank the reviewer for their comments. Below please find our response to the issues raised:
>
> > 1. Unconstrained weights being irregular
>
> The reviewer’s observation is indeed correct, and we also observe this in the “Discussion” paragraph around line 157 of the paper. In response to this issue, we have introduced a regularization scheme for the influence weights that (1) enforces the weights to sum to the size of the source dataset, (2) constrains them to be non-negative, and (3) applies L2 regularization to prevent overly large values. This leads to what we refer to as “Robust Weights” (introduced around line 166), which effectively address the concerns raised by the reviewer. An example of the resulting weight distribution is shown in Figure 2 (middle).
>
> > 2. The cost of warm-up
>
> We acknowledge this limitation, as noted in the Limitations section of our paper. However, we believe it is not a major concern for two reasons: (1) as the training pool grows, the cost of a brief warm-up on a small random subset becomes negligible compared to the cost of embedding the full dataset; and (2) the warm-up phase can be shortened (as demonstrated in Appendix A) or further compressed—for example, using Low-Rank Adaptation (LoRA) [1], as done in the LESS method [2]. A more thorough exploration of warm-up optimization is left for future work.
>
> To provide a clearer picture of the current runtime, we estimate the cost of our (unnecessarily long) warm-up phase. In the setting of Table 1, the warm-up cost for Llama2-7B is approximately 420 TeraFLOPs (TF). Including this, the total runtime of Influence Distillation with 4096 landmarks is about 1292 TF, which is still 2.2x faster than the RDS+ baseline.
>
> [1] https://arxiv.org/pdf/2106.09685
>
> [2] https://arxiv.org/pdf/2402.04333
>
> > 3. The cost of gradients and Hessian
>
> We agree that the naive version of Influence Distillation would be prohibitively expensive due to the need to compute and store full gradients and Hessians. This is why Section 4 of our submission is dedicated to addressing exactly these challenges. Specifically: (1) we approximate gradients using a theoretically grounded landmark-based method, which enables efficient and accurate estimation of gradient similarities between source and target samples; and (2) we argue that explicit computation of Hessians is unnecessary, as the second-order term becomes negligible under practical learning rates. This significantly reduces the computational burden while maintaining performance, shifting the pareto frontier of runtime vs accuracy as shown in Figure 1.

---

> > ### Author Response · Authors · 2025-08-04
> >
> > As the discussion period is coming to an end soon, we would like to kindly ask the reviewer to take a moment to review our rebuttal. We have included new results and additional discussion, and we hope we have addressed your concerns. We are happy to discuss any remaining questions or feedback.

---

> > > ### Comment · Reviewer_MsKh · 2025-08-05
> > >
> > > The author provide sufficient explanation to my questions. I will uphold my score.

---

### Official Review · Reviewer_29so · 2025-06-28

**Clarity:** 2
**Significance:** 2
**Originality:** 2
**Rating:** 4
**Confidence:** 4

**Summary:**

This paper propose a influence-based distillation method to optimal weight or select data samples. It estimates each sample's influence on a downstream target task using a second-order approximation of the loss, and accelerates computation through a landmark-based approximation combined with Jacobian-vector product (JVP) embeddings. The method supports both SGD and Adam optimizers, and introduces regularization for robust weight estimation.

**Questions:**

- What is the impact of the chosen landmark samples? Do differences in sample quality or token length affect the final performance of Influence Distillation?

- Could the authors explain why performance drops significantly when the number of landmark samples reaches 8192? Intuitively, increasing the number of landmarks should enhance performance rather than degrade it.

- Could using a different embedding model—with a potentially different embedding space—affect performance or violate the assumption made in Line 235?

- The reviewer wonders why sample weights are not used during training (Lines 309–310) and whether this is due to the approximations in Section 4. Further explanation would be appreciated.

- The reviewer recommends including the performance of the raw base model for reference.

- Typo: " fine-tuneing" in Line 287

**Ethical Concerns:**

["NO or VERY MINOR ethics concerns only"]

**Final Justification:**

Most of my concerns have been addressed. Therefore, I lean positive towards this work and raise my rating score to 4.

**Limitations:**

Yes

**Quality:**

2

**Strengths And Weaknesses:**

**Strengths**
-  This paper is straightforward and  easy to follow.
- Derives optimal sample weights for both SGD and Adam.
- Provide Landmark + JVP-based approximations reduce computational cost heavily, which is novel and interesting.

**Weaknesses**

- The method shows limited gains over baselines, especially Full training, and lacks comparisons to stronger recent methods like prompt-based scoring [1-2] or LESS, weakening its empirical support.

- The paper introduces multiple approximations (e.g., gradient projection, landmark-based gradient estimation, first-order influence), but does not analyze their joint effect on the robust Influence Distillation objective. For instance, in the experiments, the authors claim that the sample weights no longer contribute to performance, which raises concerns about the overall impact of these approximations.
 Besides, it remains unclear whether the combined approximations still preserve the intended influence estimation accuracy. The reviewer recommends including a small ablation study to analyze the correlation between the approximated influence scores and the true (non-approximated) influence scores.



[1] AlpaGasus: Training A Better Alpaca with Fewer Data, ICLR 2024.

[2] Improving Data Efficiency via Curating LLM-Driven Rating Systems, ICLR 2025.

---

> ### Author Rebuttal · Authors · 2025-07-31
>
> We thank the reviewer for their constructive feedback. Below we address the raised concerns:
>
> > 1. Discussion on the baselines and related work
>
> We would like to emphasize that, compared to training on the full dataset, Influence Distillation offers significant computational savings. For instance, in the Llama2-7B experiments, full training on 200k examples requires approximately 8400 TeraFLOPs (TF), whereas Influence Distillation with 4096 landmarks requires only 1292 TF for embedding, selection, and training on the 10k subset. This results in a 6.5x speedup while recovering the accuracy of full training across the average of six tasks (see Table 1).
>
> When compared to other baselines included in our paper, we argue that Influence Distillation is Pareto-superior—that is, it is both faster and at least as accurate on average across all tasks. This trade-off is illustrated in Figure 1.
>
> Regarding a comparison with the LESS method, please see the next point below
>
> We appreciate the reviewer’s suggestion and thank them for highlighting these related works. The two additional methods (AlpaGasus and DS2) involve evaluating each sample in the pool using one or more large-scale language models, such as ChatGPT, which results in substantially higher computational cost due to the size of the models involved. Moreover, these methods are generally model-agnostic and do not offer the same level of theoretical grounding as our approach.
>
> Finally, we thank the reviewer for this valuable comment. We will incorporate the new results and include a discussion of AlpaGasus and DS2 in the next revision of the paper.
>
> > 2. Comparison with LESS [1]
>
> We thank the reviewer for this valuable comment. As noted in the last paragraph of Appendix F (also mentioned in Section 5.1, Baselines), we have included a brief discussion of the LESS method. Specifically, as an ablation, we replaced JVP embeddings with actual projected gradients. Figure 7 (right) in the appendix shows that this yields a gradient recovery of 0.9 (in terms of average cosine similarity) with 4096 landmarks, making the approach, in this case, very similar to LESS which uses exact (projected) gradients. The last row of Table 2 shows that this variant performs on par with Influence Distillation using JVP embeddings.
>
> To provide a more complete comparison, we also ran the 10k/200k selection experiment on Llama2-7B using the LESS method (with one seed due to its high computational cost). In this setup, gradients are computed after a 10k-step warmup, projected down to 8192 dimensions, and compared to the target gradients. The top 10k examples are then selected based on similarity—following the original LESS procedure. Below we present a comparison between uniform sampling, InfDist with JVP embeddings, and LESS.
>
> |Method|MMLU|GSM8k|BBH|TyDIQA|CODEX|SQuAD|Avg|
> |------|----|-----|---|------|-----|-----|----|
> |Uniform|45.6±0.43|17.5±1.08|41.8±0.20|51.6±0.38|27.0±0.60|80.8±1.05|44.1|
> |InfDist+JVP Embedding (ours)|48.3±0.21|20.3±1.65|43.2±0.67|53.6±0.34|29.5±3.14|83.2±1.02|46.4|
> |LESS|48.2|22.7|43.4|55.8|29.1|86.0|47.5|
>
> While LESS achieves higher accuracy on average, it is significantly more computationally expensive. Our method requires approximately 872 TeraFLOPs (TFs) for embedding and selection, whereas LESS requires around 8400 TFs—approximately 10x more.
>
> We thank the reviewer again for this insightful suggestion and will include these results in the main body of the next revision.
>
> [1] https://arxiv.org/pdf/2402.04333
>
> > 3. Discussion on approximations
>
> We would like to thank the reviewer for this insightful comment. As suggested, we first include an ablation measuring the similarity between approximated and non-approximated influence weights.
>
> We note that computing the similarity between all pairs of source and target gradients without projection is computationally intractable for our data scale. Specifically, it requires $O(|S| \times |T|)$ gradient computations, as we can only store a constant number of gradients in memory. Additionally, given the high dimensionality, the source data pool must be large to yield meaningful similarities. Therefore, even for the non-approximated weights, we apply random projections to reduce the gradient dimensionality to 8192, using Rademacher projections (as in LESS).
>
> For this experiment, we consider the Llama2-7B model, the 200k Tulu V2 training dataset, 32 random MMLU target samples, and 4096 landmarks. We compute both the approximate Influence Distillation weights and the "true" second-order influence weights (with both gradients and Hessian-vector products projected to 8192 dimensions), with learning rate 2e-5, which we used for our fine-tuning experiments. We find that the two sets of weights have a Pearson correlation of around 0.7, which aligns with the theory presented in Theorem 1: while the low-rank gradient approximation may yield a weak recovery of individual gradients, their similarities with target gradients are preserved due to the high dimensional structure.
>
> As for the observation that using weights during training does not improve performance, we believe this is largely due to the regularization tuning procedure. As mentioned in the paper, we choose the regularization strength $\lambda$ such that the final solution has exactly 10k non-zero weights. In practice, many of these weights are small. In the experiment described above, approximately 20% of selected samples had weights less than 5, while another 20% had weights larger than 30, with some samples with weights as high as 160. When training with these weights, the low-weight samples contribute very little and are effectively ignored, reducing the advantage of weighting. A potential alternative would be to allow a larger number of non-zero weights and then select the top-k most influential samples, but we leave this to future work.
>
> We thank the reviewer again for raising this important point and will include these new findings and discussions in the next revision of the paper.
>
> > 4. Impact of landmark selection
>
> We repeat the experiments from the appendix Figure 7 (Left) with different landmark selection methods. Specifically, we compute the average cosine similarity of the landmark-based approximated gradients with true gradients when using JVP embeddings. We fix the number of landmarks to 4096, and compare the following landmark selection methods: (1) uniformly at random, (2) leverage score-based sampling in the JVP embeddings space, (3) longest samples, and (4) shortest samples. The results are presented in the following table (the model is Llama2-7B):
>
> | |Uniform|Leverage|Shortest|Longest|
> |--|--|--|--|--|
> |Gradient recovery|0.105|0.105|0.035|0.046|
>
> These results show that deliberately choosing longest or shortest samples significantly degrades gradient approximation quality. On the other hand, choosing landmarks uniformly at random is on par with leverage score-based sampling, as mentioned in the Hyperparameters paragraph in Section 5.1.
>
> We thank the reviewer for this feedback, and will include this study in the next revision.
>
> > 5. Performance degradation when #landmarks=8192
>
> We acknowledge that the reviewer’s observation is correct: as the number of landmarks increases, the quality of gradient recovery should improve, and consequently, the performance of Influence Distillation is expected to increase. This trend is indeed reflected in the $\textit{general}$ behavior shown in Figure 3.
>
> However, as noted in Table 1, certain tasks—such as GSM8k and Codex—show a high variability and are particularly noisy. We believe the slight performance drop observed at 8192 landmarks is likely due to this noise.
>
> > 6. Transferability of embeddings between models
>
> To investigate this point, we conducted an experiment to study how the average cosine similarity between approximated and true gradients of Llama2-7B is affected by the choice of embedding model. Specifically, we compare the performance of the following embeddings: JVP from Llama2-7B (same as the target model), JVP from Llama3.2-3B, JVP from Qwen2.5-1.5B, and JVP from Qwen2.5-3B. We fix the number of landmarks at 4096. The results are summarized in the following table:
>
> | |JVP (Llama2-7B)|JVP (Llama3.2-3B)|JVP (Qwen2.5-1.5B)|JVP (Qwen2.5-3B)|RDS+ (Llama2-7B)|
> |--|--|--|--|--|--|
> |gradient recovery|0.105|0.098|0.093|0.103|0.073|
>
> Comparing these results with the recovery achieved by RDS+, we conclude that JVP embeddings are highly transferable across models. Notably, JVP embeddings computed from Qwen2.5-3B approximate the gradients of Llama2-7B with similar quality to those obtained directly from Llama2-7B itself.
>
> We appreciate the reviewer’s constructive comment and will include these results and a discussion in the next revision of the paper.
>
> > 7. Performance of raw base models
>
> We thank the reviewer for the suggestion and will include these results for all models in the next revision. For the purpose of this rebuttal, we have computed the results for Llama2-7B, which are presented in the table below:
>
> | |MMLU|GSM8k|BBH|TyDiQA|Codex|SQuAD|
> |-|----|---|---|------|-----|-----|
> |Base Llama2-7B|39.9|3.1|39.8|17.3|18.9|10.3|
> |InfDist|48.3±0.21|20.3±1.65|43.2±0.67|53.6±0.34|29.5±3.14|83.2±1.02|
>
> As expected, without instruction tuning, the base model performs poorly on all tasks.
>
> > 8. Typo: fine-tuning
>
> We appreciate the reviewer’s attention to detail and will correct this typo in the next revision.

---

> > ### Author Response · Authors · 2025-08-04
> >
> > As the discussion period is coming to an end soon, we would like to kindly ask the reviewer to take a moment to review our rebuttal. We have included new results and additional discussion, and we hope we have addressed your concerns. We are happy to discuss any remaining questions or feedback.

---

> > ### Comment · Reviewer_29so · 2025-08-05
> >
> > Thank you for detailed responses. Most of my concerns have been addressed. I lean positive toward this work and will raise my score to 4.

---

### Official Review · Reviewer_yYDh · 2025-07-05

**Clarity:** 3
**Significance:** 2
**Originality:** 3
**Rating:** 4
**Confidence:** 4

**Summary:**

This paper introduces a learned data‐sampling strategy for instruction‐tuning large language models. At its core, the method maintains a “target task” small dataset T (e.g., a benchmark’s train/dev split) and a large generic pool S. It repeatedly estimates, via an influence‐distillation objective, how much each sample in S would improve performance on T if added to the fine‐tuning batch. Sampling weights are updated to favor high‐influence examples, and the model is fine‐tuned on these examples in turn. To keep this process scalable, the authors propose a landmark‐based approximation that reduces the cost of influence estimation. They evaluate on six downstream benchmarks, showing consistent gains over uniform or heuristic sampling with only modest extra computation.

**Questions:**

1. Have you tried transferring a sampler trained on one benchmark’s T to fine‐tuning on a different downstream task without retraining the sampling weights? If so, how does performance compare? Demonstrating such transfer would greatly strengthen claims of broad utility.

2. In settings with no small held‐out dev for T, can your method bootstrap itself—e.g., by iterative self‐training or clustering‐based pseudo-targets? If not, how critical is the quality/size of T for gains?

3. Do you foresee any modular extension of your framework to unsupervised pretraining data selection or reinforcement-learning fine-tuning? Even a pilot experiment or discussion would clarify the method’s broader applicability.

**Ethical Concerns:**

["NO or VERY MINOR ethics concerns only"]

**Final Justification:**

The authors have replied to my questions. I'll keep the original score 4 due to the limited scope of the work.

**Limitations:**

The authors note reliance on a representative target dataset and mention scalability trade-offs. However, they do not explore cases where no such dataset exists or where target and source distributions diverge drastically. I recommend adding a discussion or small experiment on cold-start scenarios and potential mitigation strategies (e.g., self-supervised proxy tasks).

**Quality:**

3

**Strengths And Weaknesses:**

**Quality (Strengths)**

- Rigorous formulation of the sampling objective via influence functions, with clear derivations.

- Includes end‐to‐end cost accounting (fine‐tuning + embedding + sampling), giving a realistic sense of overhead.

- Empirical gains are consistent across six diverse instruction‐tuning benchmarks, demonstrating robustness.


**Quality (Weaknesses)**

- Limitation to instruction‐tuning stage only; does not address data selection during pretraining or reinforcement‐learning fine‐tuning, where large‐scale unsupervised data could benefit most.

- All evaluations tie the sampling policy to *each* target benchmark’s dev set; no experiments test transfer of a learned sampler to a truly unseen task.


**Clarity (Strengths)**

- Well‐structured presentation: algorithm pseudocode, clear notation for S, T, influence scores.

- Landmark approximation is motivated and explained intuitively before formalizing.


**Significance (Strengths)**

- Addresses an important bottleneck: sampling the “right” examples during instruction tuning to maximize downstream performance per compute.

- Opens a new direction for learning to sample data based on target‐task feedback, which could complement or replace heuristic curricula.


**Significance (Weaknesses)**

- Impact is limited if users lack access to any representative target dev set (cold‐start scenario).

- Would be more significant if shown to carry over across tasks or to larger unsupervised pools in pretraining.


**Originality (Strengths)**

- Novel combination of influence estimation with a learnable sampling distribution in the instruction‐tuning context.

- Landmark‐based approximation appears to be a fresh scalability trick.

---

> ### Author Rebuttal · Authors · 2025-07-31
>
> We would like to thank the reviewer for their valuable comments. Please find the answers to your concerns below.
>
> > 1. Extension to pre-training and the no-target case
>
> Our current focus is indeed on the supervised fine-tuning phase, rather than pre-training or reinforcement learning. This choice reflects our goal of improving training efficiency in realistic fine-tuning settings, where large amounts of instruction-tuning data are available and training cost is a bottleneck.
>
> That said, we believe that several components of our method are applicable to pre-training scenarios as well:
>
> Gradient approximation via JVP embeddings remains effective in pre-training after a short warmup period, as the gradients have been shown to exhibit low-rank structure after this initial phase [3]. This suggests that the embedding-based similarity approach could be extended to the pre-training stage with minimal overhead.
>
> In cases where there is no explicit target distribution, there are two possible adaptations worth exploring:
>
> (i) One could set the source and target sets to be the same (i.e., $S=T$). In this case, both sets share a common gradient matrix $G$, which can be written as $G=CL$ (similar to the paper). This allows for efficient computation of pairwise gradient similarities as $G^TG=L^TC^TCL$. This structure could enable selection of influential samples using, for example, leverage score-based sampling.
>
> (ii) Alternatively, one could define a small, high-quality subset of examples as the target distribution and select pre-training data based on their influence relative to that set.
>
> While these directions are promising, they would require additional experiments to fully validate. We view this as an exciting avenue for future work and thank the reviewer for this suggestion.
>
> [3] https://arxiv.org/abs/2403.03507
>
>
> > 2. Transferability of weights
>
> We agree that this is an interesting experiment. To investigate this, we compute sample weights for different target sets as described in the paper, and analyze their pairwise Pearson correlations. For this analysis, we use the Llama2-7B model with 1024 landmarks and a data pool of 200k examples. As shown in the table below, the computed weights for four of the six tasks exhibit high correlation (0.5–0.8), suggesting that sample weights computed for one task may transfer well to others.
>
> |                | MMLU | GSM8k | BBH | TyDiQA | Codex  | SQuAD  |
> |----------------|------------|-------------|-----------|--------|--------|--------|
> | MMLU     | 0.9961     | -0.0430     | -0.5430   | 0.6953 | 0.7422 | 0.7734 |
> | GSM8k    | -0.0430    | 0.9961      | 0.2520    | 0.2188 | 0.2637 | -0.0938|
> | BBH      | -0.5430    | 0.2520      | 1.0000    | -0.2021| -0.2295| -0.6719|
> | TyDiQA         | 0.6953     | 0.2188      | -0.2021   | 1.0000 | 0.5938 | 0.5312 |
> | Codex          | 0.7422     | 0.2637      | -0.2295   | 0.5938 | 1.0000 | 0.5547 |
> | SQuAD          | 0.7734     | -0.0938     | -0.6719   | 0.5312 | 0.5547 | 1.0000 |
>
>
> To further assess cross-task generalization, we use the sample weights computed for MMLU to select a 10k subset from the training pool. We then train a model on this subset and evaluate it on each of the remaining tasks. The results are presented in the table below, comparing Influence Distillation and uniform sampling (the results are averaged over three seeds):
>
> | Method  | MMLU | GSM8k | BBH  | TyDIQA | CODEX | SQuAD |
> |---------|------|-------|------|--------|-------|-------|
> | Uniform | 45.6 | 17.5  | 41.8 | 51.6   | 27.0  | 80.8  |
> | InfDist | 48.2 | 16.8  | 42.6 | 50.6   | 28.2  | 82.1  |
>
> These results show that, even though the samples were selected for MMLU, Influence Distillation still outperforms uniform selection in most tasks when evaluated for three seeds.
>
> > 3. Study on size of T
>
> To study the effect of the target set size, we vary the number of samples per category in the MMLU target distribution. MMLU contains 57 categories, and in the main paper we use 5 samples per category (285 samples in total) as the target set. For this ablation, we repeat the Llama2-7B experiment using 1, 2, 3, and 4 samples per category, resulting in target set sizes of 57, 114, 171, and 228, respectively.
>
> | Target Size |57|114|171|228|285|
> |-|------|------|------|------|------|
> |MMLU accuracy|48.1|48.6|48.2|48.1|48.5|
>
> These results suggest that Influence Distillation is relatively robust to the size of the target dataset in this setting.
> We thank the reviewer for this valuable comment and will include these findings—along with similar analyses for other target datasets—in the next revision of the paper.
>
> > 4. Divergence of Target and Source
>
> In this case, it intuitively would be hard to perform well on the target distribution for which we don’t have any data. We believe that this would be a limitation of all data selection methods.

---

> > ### Author Response · Authors · 2025-08-04
> >
> > As the discussion period is coming to an end soon, we would like to kindly ask the reviewer to take a moment to review our rebuttal. We have included new results and additional discussion, and we hope we have addressed your concerns. We are happy to discuss any remaining questions or feedback.

---

> > > ### Comment · Reviewer_yYDh · 2025-08-05
> > >
> > > Good work! The supplement experiments and discussion should provide a better vision for the potential audience. I'll maintain the current positive rating.

---

### Official Review · Reviewer_nWgK · 2025-07-07

**Clarity:** 2
**Significance:** 3
**Originality:** 3
**Rating:** 4
**Confidence:** 3

**Summary:**

The paper introduces the Influence Distillation framework, which selects a subset of data from a large training corpus to improve both the performance of LLMs trained on the selected data and the overall training efficiency. The core idea of Influence Distillation is to assign weights to training samples based on their estimated influence on downstream tasks. This influence is computed using a second-order Taylor approximation of the loss on the target dataset, following a gradient step on the training data. Experimental results show that the Influence Distillation framework outperforms several existing data selection methods, while also reducing computational cost and achieving data selection up to 3.5 times faster.

**Questions:**

Please move the proofs of the formulas from the main text to the appendix.

**Ethical Concerns:**

["NO or VERY MINOR ethics concerns only"]

**Final Justification:**

4: Borderline accept

**Limitations:**

yes

**Quality:**

3

**Strengths And Weaknesses:**

> **Strengths**

1. The proposed Influence Distillation framework outperforms several strong baselines, including Mid-PPL and RDS+, in terms of model performance as well as the time and computational cost of data selection.
2. The paper introduces landmark-based gradient approximation and JVP embeddings to improve the efficiency of gradient computation, thereby reducing the computational cost of the Influence Distillation framework.

> **Weaknesses**

1. The paper does not compare its method with the LESS approach [1] for data selection, which is similar to the Influence Distillation framework. Both methods select training examples based on their influence on the validation set, with influence calculated using model gradients.
2. I wonder whether it is necessary to select as many as 10k examples for instruction tuning. When training on the full dataset, LLM performance may converge after only a few thousand examples. It might be more informative to report the performance of LLMs at various training steps on the selected datasets obtained through different data selection strategies.

**References**

[1] LESS: Selecting Influential Data for Targeted Instruction Tuning (ICML 2024)

---

> ### Author Rebuttal · Authors · 2025-07-31
>
> We thank the reviewer for their insightful feedback. Here are our answers to their questions.
>
> > 1. Comparison with LESS [1]
>
> We thank the reviewer for this valuable comment. As noted in the last paragraph of Appendix F (also mentioned in Section 5.1, Baselines), we have included a brief discussion of the LESS method. Specifically, as an ablation, we replaced JVP embeddings with actual projected gradients. Figure 7 (right) in the appendix shows that this yields a gradient recovery of 0.9 (in terms of average cosine similarity) with 4096 landmarks, making the approach, in this case, very similar to LESS which uses exact (projected) gradients. The last row of Table 2 shows that this variant performs on par with Influence Distillation using JVP embeddings.
>
> To provide a more complete comparison, we also ran the 10k/200k selection experiment on Llama2-7B using the LESS method (with one seed due to its high computational cost). In this setup, gradients are computed after a 10k-step warmup, projected down to 8192 dimensions, and compared to the target gradients. The top 10k examples are then selected based on similarity—following the original LESS procedure. Below we present a comparison between uniform sampling, InfDist with JVP embeddings, and LESS.
>
> |Method|MMLU|GSM8k|BBH|TyDIQA|CODEX|SQuAD|Avg|
> |------|----|-----|---|------|-----|-----|----|
> |Uniform|45.6±0.43|17.5±1.08|41.8±0.20|51.6±0.38|27.0±0.60|80.8±1.05|44.1|
> |InfDist+JVP Embedding (ours)|48.3±0.21|20.3±1.65|43.2±0.67|53.6±0.34|29.5±3.14|83.2±1.02|46.4|
> |LESS|48.2|22.7|43.4|55.8|29.1|86.0|47.5|
>
> While LESS achieves higher accuracy on average, it is significantly more computationally expensive. Our method requires approximately 872 TeraFLOPs (TFs) for embedding and selection, whereas LESS requires around 8400 TFs—approximately 10x more.
>
> We thank the reviewer again for this insightful suggestion and will include these results in the main body of the next revision.
>
> [1] https://arxiv.org/pdf/2402.04333
>
> > 2. Accuracy throughout training and the size of the target set
>
> We would like to note that our 10k example selection setting follows the setting introduced in [2], where 10k examples were selected from the same dataset.
>
> To be complete, we repeated the Llama2-7B experiment on MMLU using Influence Distillation, Uniform sampling, and RDS+. In this run (with a single seed), we saved model checkpoints every 25 steps to better track performance over time. Below, we report MMLU accuracies throughout training:
>
> |Method |Step 25|Step 50|Step 75|Step 100|Step 125|Step 150|
> |-------|--------|--------|--------|--------|--------|--------|
> |Uniform|44.7|45.4|45.8|46.2|46.2|46.1|
> |RDS+|45.1|45.3|46.0|45.9|46.2|46.2|
> |InfDist|45.2|46.1|47.9|48.3|48.4|48.5|
>
> We also repeated the experiment using full training on the entire dataset. Our results show that full training reaches an accuracy of 48.5 only around step 1100. In contrast, Influence Distillation achieves the same accuracy at step 150 – over 7x faster.
>
> We thank the reviewer for this insightful comment. We will include these results in the next revision.
>
> [2] https://arxiv.org/pdf/2503.01807
>
> > 3. Placement of proofs in the main body
>
> Thank you for the suggestion regarding the placement of proofs. Upon review, we note that the main text does not currently include full proofs; the closest instance is the four-line Taylor expansion leading to Equation 2. We agree that this would be better suited to the appendix and will make this adjustment in the next revision.

---

> > ### Author Response · Authors · 2025-08-04
> >
> > As the discussion period is coming to an end soon, we would like to kindly ask the reviewer to take a moment to review our rebuttal. We have included new results and additional discussion, and we hope we have addressed your concerns. We are happy to discuss any remaining questions or feedback.

---

> ### Comment · Reviewer_nWgK · 2025-08-05
>
> Thank you to the authors for their detailed responses. The additional experimental results further demonstrate the effectiveness of the proposed Influence Distillation (InfDist) framework. However, although InfDist requires significantly lower computational cost than LESS, it does not outperform LESS. **Nonetheless, I believe InfDist is a valuable contribution to the research community, and I would be pleased to see this paper accepted.** Since a rating of 4 is already considered positive, I will maintain my score.

---

> > ### Author Response · Authors · 2025-08-07
> >
> > We would like to thank the reviewer for their positive feedback and support.
> >
> > We wish to make one additional clarification regarding LESS. While LESS is a strong baseline in terms of accuracy, we argue that its computational cost is **prohibitively high** in realistic settings. Specifically, it requires computing and projecting gradients for every single sample in the pool, followed by fine-tuning on the selected subset. Even if we ignore the cost of projecting gradients, which is significant in practice, the overall cost remains higher than training on the full dataset for a full epoch. For example, the caption of Figure 1 in [1] notes: "We do not run LESS with 5.8M samples due to its high compute cost." This limitation suggests that LESS is not practically comparable to Influence Distillation, which is designed to be highly efficient via landmark-based approximation.
> >
> > We once again thank the reviewer for their valuable comments and constructive feedback, and we hope this clarification further addresses any remaining concerns.
> >
> > [1] https://arxiv.org/pdf/2503.01807

---

### Note · Authors · 2025-08-12

We sincerely thank all reviewers for their constructive feedback, which has helped improve our work and clarify its contributions. We addressed their concerns by:

**Additional supporting experiments**: we conducted new studies to further justify aspects of Influence Distillation, including: reporting accuracies for multiple checkpoints during training, assessing the transferability of Influence Distillation weights across tasks, analyzing the effect of target set size, comparing with ideal non-approximated weights, evaluating alternative methods for selecting landmarks and their count, examining transferability of JVP embeddings, and more.

**Comparison with more baselines**: we added new results comparing Influence Distillation with LESS [1]. While LESS achieves slightly higher accuracy, it is prohibitively costly in practice; 10x slower than Influence Distillation and even slower than training on the full dataset. We also addressed other related work mentioned by reviewers, clarifying their relevance and discussing their strengths and limitations.

**Extension to pre-training**: we discussed how Influence Distillation can be extended to pre-training settings and scenarios without a target dataset, outlining exciting future directions.

We truly appreciate that the reviewers engaged positively with these discussions and indicated that our detailed replies resolved their concerns and improved their evaluation of the paper.

[1] https://arxiv.org/pdf/2402.04333

---

### Decision · Program_Chairs · 2025-09-17

**Decision:**

Accept (poster)

**Comment:**

This paper studies the problem of data selection for efficient model training. The authors introduces Influence Distillation, a data selection framework that uses second-order information to assign weights to training samples. The authors also propose a landmark-based approximation where influence is compute on a small landmark subset and propagated to the remaining samples. The proposed method achieves good empirical performance compared to several baselines. Reviewers raised concerns regarding comparisons to the LESS approach [1]. In the rebuttal, the authors showed that while LESS achieves higher performance, the proposed approach is much more efficient. After the rebuttal, all reviewers leaned toward acceptance.

[1] LESS: Selecting Influential Data for Targeted Instruction Tuning (ICML 2024)